# CONTINUOUSLY AUGMENTED DISCRETE DIFFUSION MODEL FOR CATEGORICAL GENERATIVE MODELING

**Huangjie Zheng**[1], **Shansan Gong**[1,2]*, **Ruixiang Zhang**[1], **Tianrong Chen**[1], **Jiatao Gu**[1]
**Mingyuan Zhou**[1,3], **Navdeep Jaitly**[1], **Yizhe Zhang**[1]
[1]Apple    [2]Hong Kong University    [3]The University of Texas at Austin
`{huangjie_zheng, ruixiangz, tchen54, jgu32, njaitly, yizzhang}apple.com`
`hisansas@gmail.com, mingyuan.zhou@mccombs.utexas.edu`

## ABSTRACT

Standard discrete diffusion models treat all unobserved states identically by mapping them to an absorbing `[MASK]` token. This creates an "information void" where semantic information that could be inferred from unmasked tokens is lost between denoising steps. We introduce *Continuously Augmented Discrete Diffusion* (CADD), a framework that augments the discrete state space with a paired diffusion in a continuous latent space. This yields graded, gradually corrupted states in which masked tokens are represented by noisy yet informative latent vectors rather than collapsed "information voids". At each reverse step, CADD may leverage the continuous latent as a semantic hint to guide discrete denoising. The design is clean and compatible with existing discrete diffusion training. At sampling time, the strength and choice of estimator for the continuous latent vector enables a controlled trade-off between mode-coverage (generating diverse outputs) and mode-seeking (generating contextually precise outputs) behaviors. Empirically, we demonstrate CADD improves generative quality over mask-based diffusion across text generation, image synthesis, and code modeling, with consistent gains on both qualitative and quantitative metrics against strong discrete baselines. Our codebase is available at: `https://github.com/apple/ml-CADD`

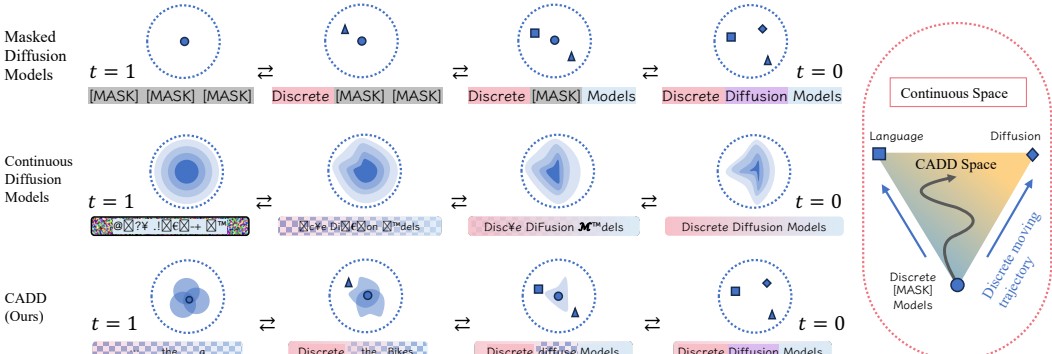

Figure 1: Comparison of diffusion models in different modeling spaces. Masked diffusion models use `[MASK]` as noise and decode through a single mask-to-token path, while continuous diffusion models explore the Gaussian space but often produce unreadable tokens due to the vast search space. CADD combines the *stability* of masked diffusion with the *flexibility* of continuous diffusion, enabling better token decoding for masked positions.

## 1 INTRODUCTION

Diffusion models have significantly advanced generative modeling tasks (Sohl-Dickstein et al., 2015; Ho et al., 2020; Song et al., 2021; Dhariwal & Nichol, 2021; Karras et al., 2022), particularly in

---

*This work was done when Shansan Gong, Mingyuan Zhou, and Navdeep Jaitly were at Apple.

image synthesis (Saharia et al., 2022; Esser et al., 2024; Polyak et al., 2024; Zheng et al., 2024a; Brooks et al., 2024). Recently, with the rapid progress on discrete diffusion models (Austin et al., 2021a; Hoogeboom et al., 2021; Lou et al., 2024), diffusion models have become a strong tool on the discrete categorical data domain, such as text generative modeling and code generation (Gat et al., 2024; Gong et al., 2023; 2025b).

Early work on Continuous Diffusion Models (CDMs) for categorical data maps tokens into a continuous space, applies Gaussian diffusion to the representations, and then rounds back to discrete symbols (Li et al., 2022; Dieleman et al., 2022; Han et al., 2022; Zhang et al., 2023; Gulrajani & Hashimoto, 2023). This route preserves smooth semantic signals and enables the use of established score-based methods. In parallel, Masked Diffusion Models (MDMs) have recently shown strong results for categorical data (Shi et al., 2024; Sahoo et al., 2024; Nie et al., 2025): instead of adding noise in an embedding space, MDMs progressively mask tokens over time and learn to unmask them, yielding clear training signals via token-level cross-entropy.

Despite their respective successes, both approaches have limitations, which illustrated in Figure 1. (i) MDMs suffer from information loss due to their use of absorbing [MASK] state (Wang et al., 2025). This design collapses all unobserved possibilities into one symbol, erasing any information about how close a corrupted position is to the original token, creating an "information void". This reduces the information available for the model to resolve ambiguity and maintain global semantic coherence. As an example shown in the right of the figure, if a masked token could plausibly be "Language" or "Diffusion", the [MASK] representation offers no semantic clue to favor one over the other, forcing the model to make a hard choice without graded guidance. (ii) While CDMs can represent semantic proximity, they face a different challenge, known as over-smoothing. Because the denoising process occurs entirely in a continuous embedding space and discretization to tokens only happens in the end (Gao et al., 2022), their continuous denoising objective can over-smooth token identities, which makes it difficult for the model to make precise predictions without localized context.

To address these challenges, we propose **Continuously Augmented Discrete Diffusion (CADD)**, which integrates the strengths of CDMs into MDMs. CADD retains the discrete masking trajectory and augments it with a paired conditional Gaussian diffusion in a continuous semantic embedding space. Our forward process jointly evolves the token sequence and its latent, so positions that are masked in the discrete path are accompanied by noisy yet informative latent vectors rather than information voids. In the reverse process, the model uses the continuous latent as a soft semantic hint to guide token denoising at each step, while the discrete context constrains the latent dynamics locally. Returning to Figure 1, the continuous manifold offers a graded path between candidates ("Language" and "Diffusion", in this case), and the discrete neighborhood restricts the search space, allowing movement within the triangular region between hypotheses and enabling smooth transitions driven by the hints. In addressing the limits of both pure MDMs and CDMs, our contributions are:

1. *Better token prediction with soft hints.* For masked positions, the continuous latent preserves graded proximity to the ground-truth token embedding, which reduces ambiguity and makes discrete prediction easier.

2. *Diversity without off-manifold drift.* At inference, one can resample the continuous latent (e.g., multiple latent draws per discrete state) to explore alternative yet valid choices for a token or span. Because the discrete head stays grounded in the vocabulary, diversity comes from semantic variation rather than uncontrolled noise.

3. *Training and sampling remain simple.* CADD keeps standard cross-entropy for tokens and a standard diffusion loss for the continuous head. The sampler can alternate or jointly update the discrete and continuous states.

4. *Efficient fine-tuning.* CADD requires no special architecture and can reuse the same backbone as any MDM, enabling efficient fine-tuning of existing MDM to gain the above benefits.

## 2 RELATED WORK

**Discrete Diffusion Models** Discrete diffusion models (Hoogeboom et al., 2021; Zheng et al., 2024b; Austin et al., 2021a) operate by defining a Markov chain over the discrete token space, gradually diffusing the data with either uniform or absorbing transitions. Later, the model was unified and simplified to continuous-time masked diffusion models (Campbell et al., 2022; Lou et al., 2024; Shi

et al., 2024; Sahoo et al., 2024; Zhang et al., 2025b). Building on this, several recent works further scaled diffusion LMs to 7B parameters (Gong et al., 2025a; Ye et al., 2025; Nie et al., 2024), achieving performance on par with AR models. Parallel efforts explored unified multimodal variants that model text and images both in discrete token (Yang et al., 2025; Li et al., 2025). However, because masked diffusion models do not allow unmasked tokens to change, errors can accumulate during generation because of suboptimal unmasking in the earlier steps. Several enhanced (re-)masking techniques have been proposed, using bits and simplex representation to enrich the binary choice of masking (Chao et al., 2025; Song et al., 2025a), remasking during the reverse process (Gat et al., 2024; Zhao et al., 2024; Wang et al., 2025), enabling edit operations (Havasi et al., 2025; Song et al., 2025b).

**Continuous Relaxations for Discrete Data**    Early continuous approaches either learn denoising in a latent embedding without explicit statistical structure (Li et al., 2022; Dieleman et al., 2022; Chen et al., 2023) or fully relax tokens into unconstrained Euclidean space as simplex (Han et al., 2022; Karimi Mahabadi et al., 2024; Tae et al., 2025; Jo & Hwang, 2024; 2025). However, such unconstrained relaxations often fail to preserve the inherent discreteness and categorical semantics of language (Gulrajani & Hashimoto, 2023). More recent methods impose structure in the logit space (Hoogeboom et al., 2021; Graves et al., 2023) or directly on the probability simplex via Dirichlet priors (Avdeyev et al., 2023; Stärk et al., 2024), enforcing stronger statistical constraints on the noising process. Flow-matching techniques further treat the simplex as a statistical manifold (Liu et al., 2023; Cheng et al., 2024; Davis et al., 2024), yet these approaches still lag behind discrete diffusion models in generation fidelity. Recently, Zhang et al. (2025a) leveraging density models with normalizing flow (Zhai et al., 2025; Gu et al., 2025) for flexible language modeling, and Sahoo et al. (2025) connect discrete diffusion language models and the underlying Gaussian diffusion.

**Bridging Through the Lens of Mode Balancing**    Our work is also motivated by balancing mode seeking and mode covering. Related efforts pursue this balance via guidance methods that tune the diversity–precision trade-off (Dhariwal & Nichol, 2021; Ho & Salimans, 2022); score-distillation approaches that sharpen samples while retaining diffusion training for coverage (Poole et al., 2022; Song et al., 2023; Luo et al., 2023; Yin et al., 2024; Zhou et al., 2024; Zhang et al., 2025b); and techniques that improve GAN mode coverage using diffusion or augmentation (Zheng & Zhou, 2021; Zheng et al., 2023a; Wang et al., 2023; Karras et al., 2020; Zhao et al., 2020). Similar effects have been observed when distilling in a paired continuous space (Sahoo et al., 2025). From this perspective, the discrete path in CADD is naturally mode-seeking, while the continuous channel spreads probability mass to cover plausible alternatives for the next token.

## 3    PRELIMINARY

Let $\boldsymbol{x}_0 = (\boldsymbol{x}_0^1, \ldots, \boldsymbol{x}_0^n)$ represent a sequence of discrete tokens in a vocabulary set $\mathcal{V} = \{1, 2, ..., V\} \cup \{\boldsymbol{m}\}$ that containing $V$ tokens plus a mask token $\boldsymbol{m}$ (`[MASK]`), i.e., for any positions $i$, $\boldsymbol{x}_0^i \in \{0, 1\}^{V+1}$ is a one-hot vector. Let $\boldsymbol{w}_\theta : \mathcal{V} \to \mathbb{R}^d$ be a learnable token embedding matrix and the embedding latent representations are deterministically transformed as $\boldsymbol{z}_0 := \boldsymbol{w}_\theta(\boldsymbol{x}_0)$, and $\boldsymbol{z}_0 \in \mathbb{R}^{n \times d}$.

**Discrete Diffusion Models**    The forward diffusion process is performed through an element-wise conditional sampler $q(\boldsymbol{x}_t | \boldsymbol{x}_0) = \prod_{i=1}^{n} q(\boldsymbol{x}_t^i | \boldsymbol{x}_0^i)$, defined as ($\delta(\cdot)$ denotes the dirac function):

$$q(\boldsymbol{x}_t^i | \boldsymbol{x}_0^i) \triangleq \alpha_t \delta(\boldsymbol{x}_t^i - \boldsymbol{x}_0^i) + (1 - \alpha_t) \delta(\boldsymbol{x}_t^i - \boldsymbol{m}), \tag{1}$$

where $\alpha_t \in [0, 1]$ is a strictly decreasing scheduling function following $\alpha_t = \prod_{s=1}^{t}(1 - \beta_s)$. The reverse process aims to learn $p(\boldsymbol{x}_s | \boldsymbol{x}_t)$ for $0 \leq s < t \leq 1$. This is typically achieved by training a model $p_\theta(\boldsymbol{x}_0 | \boldsymbol{x}_t)$ to predict the original data from a corrupted state, optimized by minimizing a variational bound on the negative log-likelihood, denoting $\alpha_t'$ the derivative of $\alpha_t$ *w.r.t.* $t$:

$$\mathcal{L}_{\text{vb}}(\boldsymbol{x}_0; \theta) \triangleq \mathbb{E}_{t, \boldsymbol{x}_t \sim q(\cdot | \boldsymbol{x}_0)} \left[ -\frac{\alpha_t'}{1 - \alpha_t} \log p_\theta(\mathbf{x}_0 | \mathbf{x}_t) \right]. \tag{2}$$

**Continuous Diffusion Models**    Continuous diffusion models corrupt real-valued data $\boldsymbol{z}_0 \in \mathbb{R}^{n \times d}$ by adding Gaussian noise scheduled by $\bar{\gamma}_t$. The forward process $q(\boldsymbol{z}_t | \boldsymbol{z}_0)$ is a Gaussian distribution with a closed form:

$$q(\boldsymbol{z}_t | \boldsymbol{z}_0) = \mathcal{N}(\boldsymbol{z}_t; \sqrt{\bar{\gamma}_t} \boldsymbol{z}_0, (1 - \bar{\gamma}_t) \boldsymbol{I}) \tag{3}$$

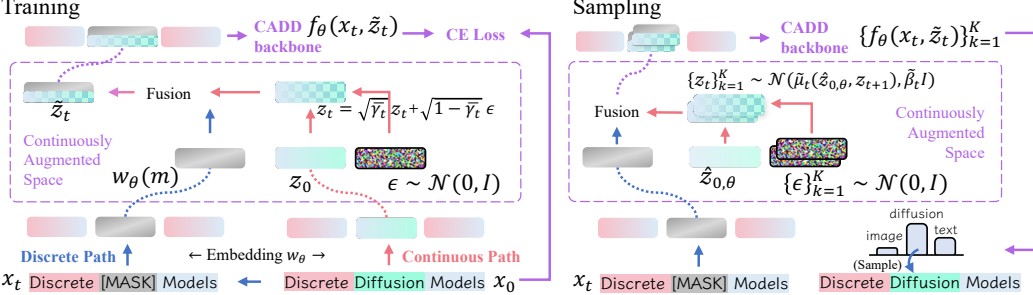

Figure 2: Illustrative depiction of CADD model, combining both the discrete and continuous feature of the data. In training, the clean token at the masked position will be created by embedding matrix and used to form the noisy embedding according to the continuous forward. In sampling, the model is able to predict a diverse distribution of possible tokens by sampling multiple $z_{t+1}$. Then the predicted tokens will be recycled into the embedding matrix to form $\hat{z}_{0,\theta}$ for the next iteration.

where $\bar{\gamma}_t$ is a noise schedule analogous to $\alpha_t$, with $\bar{\gamma}_t = \prod_{s=1}^{t} \gamma_s$ holding. The reverse process $p_\theta(z_{t-1}|z_t)$ is trained by fitting a network $f_\theta(\cdot)$ with a MSE objective reweighted by signal-to-noise ratio (SNR) function $\lambda(\bar{\gamma}_t, t)$:

$$\mathcal{L}_{\text{vb}}(z_0; \theta) \triangleq \mathbb{E}_{t, \mathbf{x}_t \sim q(\cdot|z_0)} \left[ \lambda(\bar{\gamma}_t, t) \| f_\theta(z_t; t) - z_0 \|^2 \right]. \tag{4}$$

## 4 CONTINUOUSLY AUGMENTED DISCRETE DIFFUSION (CADD)

Here we introduce Continuously Augmented Discrete Diffusion (CADD). The high-level intuition is to mitigate the sudden information loss that occurs when tokens are replaced by an absorbing state in discrete diffusion. Inspired by the smooth signal degradation in Gaussian diffusion, CADD augments the discrete state space with a continuous latent variable, $z_t$. This variable is paired with discrete tokens $x_t$ and is designed to retain semantics of a token's original signal even when tokens in $x_t$ are masked. Guided by a set of latent vectors $\{z_t^{(k)}\}_{k=1}^{K}$, the model predicts next tokens by:

$$p_\theta(x_{t-1} \mid x_t) = \mathbb{E}_{z_t}[p_\theta(x_{t-1} \mid x_t, z_t)] \approx \sum_{k=1}^{K} p_\theta(x_{t-1} \mid x_t, z_t^{(k)}). \tag{5}$$

Conditioning continuous view of the underlying content at step $t$ and traverse on the $z_t$ space, the expectation averages over plausible continuous states so the predictor could realize the distribution of the possible tokens more accurately. The full model design is illustrated in Figure 2 and we present the detailed designs in the following sections. Noted that although we use continuous-time notation $s$ and $t$ for diffusion steps, to improve readability, we also denote specific consecutive steps in the diffusion process by $t$ and $t-1$, with total $T$ steps. Below we present the construction of CADD with main derivations. For more detailed ELBO derivations and proofs, please refer to Appendix A.

### 4.1 FORWARD

To let $z_t$ retain semantic hints of tokens in $x_t$ when they are masked, we define the joint transition:

$$q(x_t, z_t \mid x_{t-1}, z_{t-1}, x_0) := \underbrace{q(x_t \mid x_{t-1})}_{\text{discrete part}} \cdot \underbrace{q(z_t \mid z_{t-1}, x_{t-1}, x_t, x_0)}_{\text{continuous part}}, \tag{6}$$

Given a fixed discrete schedule $\{\beta_t\}_{t=1}^{T} \in [0,1)^T$ and continuous diffusion schedule $\{\gamma_t\}_{t=1}^{T}$, the forward transition of discrete and continuous part can be written as following:

$$q(x_t \mid x_{t-1}) = \prod_{i=1}^{n} \text{Categorical}(x_t^i; Q_t^\top x_{t-1}^i), \quad Q_t = (1-\beta_t)I + \beta_t \mathbf{1}\, m^\top. \tag{7}$$

$$q(z_t \mid z_{t-1}, x_{t-1}, x_t, x_0) = \prod_{i=1}^{n} \begin{cases} \delta(z_t^i - z_{t-1}^i), & x_t^i \neq m, \\ \mathcal{N}(z_t^i; \sqrt{\bar{\gamma}_t^i}\, z_{t-1}^i, (1-\bar{\gamma}_t)I_d), & x_t^i = m, x_{t-1}^i \neq m, \\ \mathcal{N}(z_t^i; \sqrt{\gamma_t}\, z_{t-1}^i, (1-\gamma_t)I_d), & x_t^i = m, x_{t-1}^i = m. \end{cases} \tag{8}$$

The discrete transition is the same as normal discrete diffusion like Austin et al. (2021a) and acts as a trigger for the continuous embedding's evolution. The continuous trajectory for an embedding remains dormant as long as its token is unmasked, holding its value constant at its original state $(\delta(\boldsymbol{z}_t^i - \boldsymbol{z}_{t-1}^i) = \delta(\boldsymbol{z}_t^i - \boldsymbol{z}_0^i)$ if $\boldsymbol{x}_t^i$ is never masked as the information is not changed). The moment a token is masked, it triggers the continuous diffusion process for its embedding. The embedding then begins a smooth degradation path determined by the Gaussian diffusion (Ho et al., 2020). If a token stays masked, its embedding simply continues along this path, becoming progressively noisier.

Now we extend the case to the marginals at timestep $t$ with the following proposition.

**Proposition 1** (Timestep-$t$ joint marginal factorization). *The marginal at timestep $t$ can be factorized:*

$$q(\boldsymbol{x}_t, \boldsymbol{z}_t \mid \boldsymbol{x}_0) = q(\boldsymbol{x}_t \mid \boldsymbol{x}_0) \cdot q(\boldsymbol{z}_t \mid \boldsymbol{x}_t, \boldsymbol{x}_0) \tag{9}$$

*Given $\alpha_t := \prod_{s=1}^t (1 - \beta_s)$ and $\overline{\boldsymbol{Q}}_t := \prod_{s=1}^t \boldsymbol{Q}_s = \alpha_t \boldsymbol{I} + (1 - \alpha_t)\, \mathbf{1}\, \boldsymbol{m}^\top$ and let $\bar{\gamma}_t := \prod_{s=1}^t \gamma_s$, with $\boldsymbol{z}_0^i = \boldsymbol{w}_\theta(\boldsymbol{x}_0^i)$, the two terms factorized above represent the discrete and continuous part:*

$$q(\boldsymbol{x}_t \mid \boldsymbol{x}_0) = \prod_{i=1}^n q(\boldsymbol{x}_t^i \mid \boldsymbol{x}_0^i), \quad q(\boldsymbol{x}_t^i \mid \boldsymbol{x}_0^i) = \mathrm{Categorical}(\boldsymbol{x}_t^i;\, \overline{\boldsymbol{Q}}_t^\top \boldsymbol{x}_0^i). \tag{10}$$

$$q(\boldsymbol{z}_t \mid \boldsymbol{x}_t, \boldsymbol{x}_0) = \prod_{i=1}^n q(\boldsymbol{z}_t^i \mid \boldsymbol{x}_t^i, \boldsymbol{x}_0^i) = \prod_{i=1}^n \begin{cases} \delta(\boldsymbol{z}_t^i - \boldsymbol{z}_0^i), & \boldsymbol{x}_t^i = \boldsymbol{x}_0^i, \\ \mathcal{N}(\boldsymbol{z}_t^i;\, \sqrt{\bar{\gamma}_t}\, \boldsymbol{z}_0^i,\, (1 - \bar{\gamma}_t)\boldsymbol{I}_d), & \boldsymbol{x}_t^i = \boldsymbol{m}, \end{cases} \tag{11}$$

A key property of the marginal distribution $q(\boldsymbol{x}_t, \boldsymbol{z}_t \mid \boldsymbol{x}_0)$ is that it conveniently factorizes into discrete and continuous components: $q(\boldsymbol{x}_t \mid \boldsymbol{x}_0)$ and $q(\boldsymbol{z}_t \mid \boldsymbol{x}_t, \boldsymbol{x}_0)$. This factorization is highly advantageous, as the distribution for each component is tractable and can be computed in closed form according to the predefined diffusion schedule.

## 4.2 REVERSE

Following Kingma et al. (2021); Xiao et al. (2022); Zhou et al. (2023), we choose the conditional distribution parameterized with neural network $f_\theta(\cdot)$ to define:

$$p_\theta(\boldsymbol{x}_{t-1}, \boldsymbol{z}_{t-1} \mid \boldsymbol{x}_t, \boldsymbol{z}_t) := q(\boldsymbol{x}_{t-1}, \boldsymbol{z}_{t-1} \mid \boldsymbol{x}_t, \boldsymbol{z}_t, \boldsymbol{x}_0 = \hat{\boldsymbol{x}}_0), \tag{12}$$

$$p_\theta(\hat{\boldsymbol{x}}_0 \mid \boldsymbol{x}_t, \boldsymbol{z}_t) = \mathrm{Categorical}\big(\mathrm{logits} = f_\theta(\boldsymbol{x}_t, \boldsymbol{z}_t)\big) \text{ if } \boldsymbol{x}_t = \boldsymbol{m} \text{ else } \delta(\hat{\boldsymbol{x}}_0 - \boldsymbol{x}_t). \tag{13}$$

The objective is to close the gap between the defined parametric distribution and the true posterior. Below we presenet the close form of the posterior. For notation simplicity, below we discuss on per position formulation and omit the notation $i$, since all distributions factorize across positions $i \in \{1, \dots, n\}$.

**Proposition 2** (Factorization of the true posterior). *By the forward construction, the posterior can be factorized in the following form*

$$q(\boldsymbol{x}_{t-1}, \boldsymbol{z}_{t-1} \mid \boldsymbol{x}_t, \boldsymbol{z}_t, \boldsymbol{x}_0) = \underbrace{q(\boldsymbol{x}_{t-1} \mid \boldsymbol{x}_t, \boldsymbol{x}_0)}_{\textit{discrete part}} \cdot \underbrace{q(\boldsymbol{z}_{t-1} \mid \boldsymbol{x}_t, \boldsymbol{z}_t, \boldsymbol{x}_{t-1}, \boldsymbol{x}_0)}_{\textit{continuous part}}. \tag{14}$$

*Moreover, we can write the close form of each component:*

$$q(\boldsymbol{x}_{t-1} \mid \boldsymbol{x}_t, \boldsymbol{x}_0) = \frac{q(\boldsymbol{x}_t \mid \boldsymbol{x}_{t-1})q(\boldsymbol{x}_{t-1} \mid \boldsymbol{x}_0)}{q(\boldsymbol{x}_t \mid \boldsymbol{x}_0)} = \begin{cases} \frac{\alpha_{t-1} - \alpha_t}{1 - \alpha_t} \boldsymbol{x}_{t-1}^\top \boldsymbol{x}_0 & \boldsymbol{x}_{t-1} \neq \boldsymbol{m}, \boldsymbol{x}_t = \boldsymbol{m} \\ \frac{1 - \alpha_{t-1}}{1 - \alpha_t} & \boldsymbol{x}_{t-1} = \boldsymbol{m}, \boldsymbol{x}_t = \boldsymbol{m} \\ \boldsymbol{x}_{t-1}^\top \boldsymbol{x}_t & \boldsymbol{x}_t \neq \boldsymbol{m}. \end{cases} \tag{15}$$

$$q(\boldsymbol{z}_{t-1} \mid \boldsymbol{x}_t, \boldsymbol{z}_t, \boldsymbol{x}_{t-1}, \boldsymbol{x}_0) = \begin{cases} \delta(\boldsymbol{z}_{t-1} - \boldsymbol{z}_0), & \boldsymbol{x}_t = \boldsymbol{x}_0 \;\textit{(no mask at t)}, \\ \delta(\boldsymbol{z}_{t-1} - \boldsymbol{z}_0), & \boldsymbol{x}_t = \boldsymbol{m}, \; \boldsymbol{x}_{t-1} = \boldsymbol{x}_0 \;\textit{(first unmask at t)}, \\ \mathcal{N}(\boldsymbol{z}_{t-1};\, \tilde{\boldsymbol{\mu}}_t,\, \tilde{\beta}_t \boldsymbol{I}_d), & \boldsymbol{x}_t = \boldsymbol{m}, \; \boldsymbol{x}_{t-1} = \boldsymbol{m}, \end{cases} \tag{16}$$

*with the following paramters:*

$$\tilde{\beta}_t = \frac{(1 - \bar{\gamma}_{t-1})(1 - \gamma_t)}{1 - \bar{\gamma}_t}, \qquad \tilde{\boldsymbol{\mu}}_t = \frac{\sqrt{\bar{\gamma}_{t-1}}(1 - \gamma_t)}{1 - \bar{\gamma}_t} \boldsymbol{z}_0 + \frac{\sqrt{\gamma_t}(1 - \bar{\gamma}_{t-1})}{1 - \bar{\gamma}_t} \boldsymbol{z}_t. \tag{17}$$

**Algorithm 1** Training of CADD

1: **Input:** data minibatch $\{\boldsymbol{x}_0^{(j)}\}_{j=1}^B$, network $f_\theta(\cdot)$, masking schedule $\{\alpha_t\}_{t=1}^T$, continuous schedule $\{\bar{\gamma}_t\}_{t=1}^T$
2: **for** $j = 1, \dots, B$ **do**
3:     draw $t_j \sim \mathrm{Uniform}(1, \dots, T)$ for each sample
4:     mask out each token position $\boldsymbol{x}_0^{(j),i}$ with probability $1 - \alpha_{t_j}$ to obtain $\boldsymbol{x}_{t_j}^{(j)}$
5:     form embeddings $\boldsymbol{z}_{\mathrm{disc}}^{(j)} \leftarrow \boldsymbol{w}_\theta(\boldsymbol{x}_{t_j}^{(j)})$, $\boldsymbol{z}_t^{(j)} \leftarrow \boldsymbol{w}_\theta(\boldsymbol{x}_0^{(j)})$
6:
7:     **for** position $i \in \{1, \dots, n\}$, if $\boldsymbol{x}_{t_j}^{(j),i} = \boldsymbol{m}$ **do**, $\boldsymbol{z}_{t_j}^{(j),i} \leftarrow \sqrt{\bar{\gamma}_{t_j}} \boldsymbol{z}_{t_j}^{(j),i} + \sqrt{1 - \bar{\gamma}_{t_j}} \boldsymbol{\epsilon}, \boldsymbol{\epsilon} \sim (\mathbf{0}, \boldsymbol{I})$
8:     **end for**
9:     $\tilde{\boldsymbol{z}}_{t_j}^{(j)} \leftarrow \boldsymbol{z}_{\mathrm{disc}}^{(j)} + \boldsymbol{z}_t^{(j)}$, compute logits $f_\theta(\tilde{\boldsymbol{z}}_{t_j}^{(j)})$
10:     optimize with cross entropy loss in Eq. (20)
11: **end for**

**Algorithm 2** Sampling of CADD

1: **Input:** desired minibatch size $B$, network $f_\theta(\cdot)$, schedules $\{\alpha_t\}_{t=1}^T$, $\{\bar{\gamma}_t\}_{t=1}^T$,
2: **for** $j = 1, \dots, B$ **do**
3:     **init:** $\boldsymbol{x}_T^{(j)} \leftarrow (\boldsymbol{m}, \dots \boldsymbol{m})$, $\boldsymbol{z}_T^{(j)} \overset{\text{i.i.d.}}{\sim} \mathcal{N}(\mathbf{0}, \boldsymbol{I})$
4:     **for** $t = T, \dots, 1$ **do**
5:         **for** $i = 1, \dots, n$, if $\boldsymbol{x}_t^{(j),i} = \boldsymbol{m}$ **do**
6:         compute $\rho_t^{\mathrm{flip}}$ and $\rho_t^{\mathrm{keep}}$ (Eq. (38))
7:         determine whether to unmask $\boldsymbol{x}_{t-1}^{(j),i} \sim \mathrm{Cat}(\rho_t^{\mathrm{flip}} f_\theta(\boldsymbol{x}_t^{(j),i}, \boldsymbol{z}_t^{(j),i}) + \rho_t^{\mathrm{keep}} \boldsymbol{m})$
8:
9:         **if** $\boldsymbol{x}_{t-1}^{(j),i} \leftarrow \boldsymbol{m}$ **then** draw $\boldsymbol{z}_{t-1}^i \sim \mathcal{N}\big(\tilde{\boldsymbol{\mu}}_t(\hat{\boldsymbol{z}}_{0,\theta}^i, \boldsymbol{z}_t^i), \tilde{\beta}_t \boldsymbol{I}_d\big)$ with Eq. (21)
10:         **else** $\boldsymbol{z}_{t-1}^{(j),i} \leftarrow \boldsymbol{w}_\theta(\boldsymbol{x}_{t-1}^{(j),i})$
11:         **end if**
12:     **end for**
13:     **end for**
14: **end for**

**Lemma 1.** *For the unmasked positions ($\boldsymbol{x}_t \neq \boldsymbol{m}$), the KL is identically $0$, and the masked positions splits exactly as*

$$D_{\mathrm{KL}}\big(q(\cdot \mid \boldsymbol{x}_t, \boldsymbol{z}_t, \boldsymbol{x}_0) \,\|\, p_\theta(\cdot \mid \boldsymbol{x}_t, \boldsymbol{z}_t)\big) = \underbrace{\rho_t^{\mathrm{flip}} \big[ -\log p_\theta(\boldsymbol{x}_0 | \boldsymbol{x}_t, \boldsymbol{z}_t) \big]}_{discrete} + \underbrace{\rho_t^{\mathrm{keep}} \mathcal{D}_{\mathrm{KL}}{}^{\mathrm{cont}}}_{continuous}, \quad (18)$$

*with the ratio that determines whether the position is going to be flipped to unmask or keep moving in the continuous space:*

$$\rho_t^{\mathrm{keep}} = \frac{1 - \alpha_{t-1}}{1 - \alpha_t}, \qquad \rho_t^{\mathrm{flip}} = \frac{\alpha_{t-1} \beta_t}{1 - \alpha_t} = \frac{\alpha_{t-1} - \alpha_t}{1 - \alpha_t}.$$

*The KL divergence in the continuous space has a reweighted MSE form:*

$$D_{\mathrm{KL}}^{\mathrm{cont}} = \frac{1}{2\tilde{\beta}_t} \big\| \tilde{\boldsymbol{\mu}}_t(\boldsymbol{z}_0, \boldsymbol{z}_t) - \tilde{\boldsymbol{\mu}}_t(\hat{\boldsymbol{z}}_{0,\theta}, \boldsymbol{z}_t^i) \big\|^2 = \frac{a_t^2}{2\tilde{\beta}_t} \| \boldsymbol{z}_0 - \hat{\boldsymbol{z}}_{0,\theta} \|^2; \ a_t = \frac{\sqrt{\bar{\gamma}_{t-1}}(1 - \gamma_t)}{1 - \bar{\gamma}_t}. \quad (19)$$

### 4.3 ALGORITHM AND IMPLEMENTATION

**Training Loss** According to Eq. (18), the model aims to learn to maximize the likelihood of discrete path, and also minimize the reweighted MSE in Eq. (19). Inspired by continuous diffusion models that used for categorical modeling, e.g., CDCD (Dieleman et al., 2022) and Plaid (Gulrajani & Hashimoto, 2023), we may estimate $\hat{\boldsymbol{z}}_{0,\theta} := \sum_v p_\theta(\hat{\boldsymbol{x}}_0 = v \mid \boldsymbol{x}_t, \boldsymbol{z}_t) \boldsymbol{w}_{\theta,v}$ and just train the model to predict correct categorical output to minimize the KL divergence. Thus, we choose to train CADD by minimizing a simple cross entropy loss as following and the training is summarized in Algorithm 1:

$$\mathcal{L}_{\mathrm{CADD}} = \mathbb{E}_t \mathbb{E}_{q(\boldsymbol{x}_t, \boldsymbol{z}_t | \boldsymbol{x}_0)} \Big[ - \sum_{i: \boldsymbol{x}_t^i = \boldsymbol{m}} \log p_\theta(\boldsymbol{x}_0^i \mid \boldsymbol{x}_t^i, \boldsymbol{z}_t^i) \Big] \quad (20)$$

Note that we may add the MSE loss in Eq. (19) to the above objective to more accurately estimate the exact variational lower bound. Empirically we find the simplified loss is more computationally efficient, thus we choose to use this loss for most of our experiments unless otherwise specified.

**Sampling** The sampling start from the last timestep $T$ of the diffusion chain. Under the absorbing forward, $\alpha_T \approx 0$, hence $p(\boldsymbol{x}_T) = \delta_{\boldsymbol{x}_T = \boldsymbol{m}}$, i.e., all tokens are masked. Since all positions are masked at $T$, the continuous prior is $p(\boldsymbol{z}_T | \boldsymbol{x}_T) = \prod_{i=1}^n \mathcal{N}(\boldsymbol{z}_T^i; \mathbf{0}, \boldsymbol{I}_d)$, which matches the forward marginal at $T$. For each timestep, given $(\boldsymbol{x}_t, \boldsymbol{z}_t)$, the network predicts

$$\pi_{\theta,i}(v) := \frac{1}{K} \sum_{k=1}^K p_\theta(\hat{\boldsymbol{x}}_0^i = v \mid \boldsymbol{x}_t, \boldsymbol{z}_t^{(k)}) \in \Delta^{V-1} \qquad \text{for each position } i.$$

If the position $i$ is unmasked, the absorbing chain keeps $\boldsymbol{x}_{t-1}^i = \boldsymbol{x}_t^i$ almost surely and the continuous variable is deterministic $\boldsymbol{z}_{t-1}^i = \boldsymbol{z}_t^i = \boldsymbol{w}_\theta(\boldsymbol{x}_t^i)$. If this position is masked, it draws a clean token $v \sim \pi_{\theta,i}(\cdot)$ with probability $\frac{1-\alpha_{t-1}}{1-\alpha_t}$ to unmask it, and the continuous latent $\boldsymbol{z}_{t-1}^i \leftarrow w_{\theta,v}$. If it remains unmasked, $\boldsymbol{z}_{t-1}^i$ moves along the continuous diffusion trajectory $\boldsymbol{z}_{t-1}^i \sim \mathcal{N}\big(\tilde{\boldsymbol{\mu}}_t(\hat{\boldsymbol{z}}_{0,\theta}^i, \boldsymbol{z}_t^i), \tilde{\beta}_t \boldsymbol{I}_d\big)$. The full sampling process in shown in Algorithm 2. Note the choice of $\hat{\boldsymbol{z}}_{0,\theta}^i$ has two options:

$$\textbf{hard:}\ \hat{\boldsymbol{x}}_0 = \arg\max_v \pi_{\theta,i}(v),\ \hat{\boldsymbol{z}}_0 = \boldsymbol{w}_\theta(\hat{\boldsymbol{x}}_0)\quad \textbf{soft:}\ \hat{\boldsymbol{z}}_{0,\theta} := \sum_v p_\theta(\hat{\boldsymbol{x}}_0 = v \mid \boldsymbol{x}_t, \boldsymbol{z}_t)\, \boldsymbol{w}_{\theta,v}. \quad (21)$$

These two choices are both valid to use depending on whether we are looking for mode-covering or mode-seeking behavior, i.e., better context localization or better diversity, respectively. In our main experiments we keep the hard option, and our empirical exploration in Appendix C.3 justify these two choices could meet the demand of these two behavior. Moreover, although CADD may leverage multi-sample for the $\boldsymbol{x}_0$ distribution estimation, for fair comparison with baselines, we keep $K = 1$ for most of our experiments. More detailed studies are also shown in the Appendix C.3.

**Implementation**    We follow the common-used design of the model architecture to let $f_\theta(\cdot)$ predict logits for categorical distribution. The discrete path follows earlier masked-diffusion setups: starting from $\boldsymbol{x}_0$, we mask a subset of positions to obtain $\boldsymbol{x}_t$, embed the mixed sequence with the learnable table and form $\boldsymbol{z}_{\text{disc}} = \boldsymbol{w}_\theta(\boldsymbol{x}_t)$. The only difference is the model needs to take an additional variable $\boldsymbol{z}_t$ input for the continuous embeddings. To achieve this, we first form the clean embeddings $\boldsymbol{z}_0 = \boldsymbol{w}_\theta(\boldsymbol{x}_0)$, and then apply noise only at masked positions using the forward marginal Eq. (11) to obtain $\boldsymbol{z}_t$. We fuse $\boldsymbol{z}_{\text{disc}}$ and $\boldsymbol{z}_t$ by element-wise addition $\tilde{\boldsymbol{z}}_t := \boldsymbol{z}_{\text{disc}} + \boldsymbol{z}_t$, and feed $\tilde{\boldsymbol{z}}_t$ to the backbone $f_\theta$ to produce per-position logits.

## 5    EXPERIMENTS

In this section we present experiments to validate the proposed CADD model through experiments on text, image, and code generation benchmarks. The evaluations are designed to assess the model's performance across diverse data modalities and scales.

### 5.1    TEXT GENERATION

**Experiment setting**    For text generation, we strictly follow the experimental setup of the Masked Diffusion Language Model (MDLM) (Sahoo et al., 2024), a common configuration for this task. We train our CADD models on the OpenWebText (OWT) dataset (Gokaslan & Cohen, 2019). Data is tokenized using the GPT-2 tokenizer with a vocabulary size of $|\mathcal{V}| = 50,257$ (Radford et al., 2019), and sequences are fixed to a length of $n = 1,024$. To be consistent with the baselines, we use a Discrete DiT backbone (Peebles & Xie, 2023) with approximately 168M parameters, and train with same number of iterations. All training hyper-parameters are identical to those in MDLM.

**Evaluation.**    We mainly compare the performance with discrete diffusion baselines in terms of the generative quality, and our evaluation protocol strictly follows that of Wang et al. (2025). We compare the performance against discrete diffusion baselines using two metrics: the MAUVE score (higher is better) (Liu et al., 2021; Pillutla et al., 2021) and generative perplexity (lower is better) (Lou et al., 2024). Further details on the evaluation setup are located in Appendix B.

**Main Results.**    Figure 3 presents the results for unconditional text generation on the OpenWebText (OWT) dataset, comparing CADD with SEDD (absorb) and MDLM across a range of sampling steps $T \in \{128, 256, 512, 1024, 4096\}$. Within the range $T \leq 1024$, all models shows improvement as the number of sampling steps increases. We can notice CADD demonstrates stronger and consistent gains as steps increase compared to SEDD and MDLM in terms of both metrics. Plotting the x-axis on a $\log_2$ scale reveals that the performance trend is approximately linear.

Extending the sampling process to $T = 4096$ further demonstrates CADD's scaling capabilities at inference time, as it continues to improve while the masked-only baselines stagnate or degrade. From $T = 1024$ to 4096, CADD's MAUVE score still increases by 0.3, and its generative perplexity

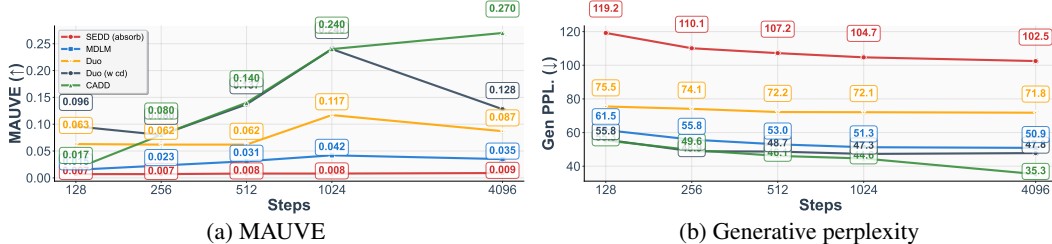

| (a) MAUVE | (b) Generative perplexity |

Figure 3: Unconditional text generative evaluation of model trained on OpenWebText (OWT) data. All method are evaluated with 128, 256, 512 1024, and 4096 sampling steps. MAUVE (higher is better) and generative perplexity (measured using GPT2-Large, lower is better) are reported.

Table 1: FID and IS evaluation on CIFAR-10. The arrow symbols denotes lower/higher is better respectively. Baseline results are quoted from Chao et al. (2025).

| Method | FID ($\downarrow$) | IS ($\uparrow$) |
|---|---|---|
| CADD (NFE=512) | **2.88** | **10.04** |
| **Discrete** | | |
| MDM (NFE=512) | 4.66 | 9.09 |
| MDM-Mixture (NFE=512) | 4.80 | 9.22 |
| MDM-Prime (NFE=512) | 3.26 | 9.67 |
| D3PM Absorb (NFE=1,000) | 30.97 | 6.78 |
| D3PM Gauss. (NFE=1,000) | 7.34 | 8.56 |
| CTDD-DG (NFE=1,000) | 7.86 | 8.91 |
| Tau-LDR (NFE=1,000) | 3.74 | 9.49 |
| Discrete FM (NFE=1,024) | 3.63 | - |
| **Continuous** | | |
| Continuous FM | 6.35 | - |
| Bit Diffusion | 3.48 | - |
| StyleGAN+ADA | 3.26 | 9.74 |
| DDPM | 3.17 | 9.46 |

Table 2: FID evaluation using model unconditionally trained on ImageNet ($32 \times 32$ resolution).

| Method | FID ($\downarrow$) |
|---|---|
| CADD (NFE=1,024) | **3.74** |
| **Discrete** | |
| MDM (NFE=1,024) | 7.91 |
| MDM-Mixture (NFE=1,024) | 8.08 |
| MDM-Prime (NFE=1,024) | 6.98 |
| **Continuous** | |
| NDM | 17.02 |
| DDPM | 16.18 |
| MSGAN | 12.30 |
| i-DODE (SP) | 10.31 |
| i-DODE (VP) | 9.09 |
| Stochastic Interp. | 8.49 |
| Soft Trunc. DDPM | 8.42 |
| ScoreFlow (subVP) | 8.87 |
| ScoreFlow (VP) | 8.34 |
| Continuous FM | 5.02 |

is scored from 44.6 to 35.3. MDLM's performance slightly worsens, which is consistent with the observation that mask-only diffusion models scale poorly with $T$ (Wang et al., 2025). Overall, CADD consistently show performance gain across all tested number of sampling steps over the mask-only discrete diffusion models, validating the effectiveness of the proposed continuous-augmented space.

## 5.2 IMAGE GENERATION

We train and evaluate our models on the CIFAR-10 (Krizhevsky et al., 2009) and ImageNet (Krizhevsky et al., 2017) datasets (resolution $32 \times 32$). For both, input images are in RGB channels, thus a dimensionality of $n = 32 \times 32 \times 3$ with $|\mathcal{V}| = 256$ pixel values per channel. For fair comparison the MDM baselines, our model architecture follows the one used in Chao et al. (2025); Gat et al. (2024), which is based on the ADM (Dhariwal & Nichol, 2021) architecture. We choose MDM-Prime (Chao et al., 2025) and its variants as our main discrete diffusion baseline. We also include its discrete and continuous diffusion model baselines for comparison (Shih et al., 2022; Ho et al., 2020; Song et al., 2021; Austin et al., 2021a; Campbell et al., 2022; Gat et al., 2024; Nisonoff et al., 2025; Lipman et al., 2022; Chen et al., 2023; Bartosh et al., 2023; Tran et al., 2019; Zheng et al., 2023b; Albergo & Vanden-Eijnden, 2023; Kim et al., 2022). To assess sample quality, we report Fréchet Inception Distance (FID) and Inception Score (IS), computed with 50,000 randomly sampled images.

We follow MDM variants to unconditionally sample images with same number of function evaluation (NFE) and report results on CIFAR-10 in Table 1. With the same NFE, we can observe CADD improves upon MDMs by a significant margin. Attaining an FID of 2.88 and an Inception Score of

Table 3: Benchmark coding capacities of AR and Diffusion LLMs in 7/8B scale. We follow the evaluation settings in DiffuCoder (Gong et al., 2025b), where EvalPlus is computed as the average of HE+ and MBPP+. The best performance in AR and Diffusion LLMs are marked in bold.

| Model | HumanEval | | MBPP | | EvalPlus | BigCodeBench (C) | | Avg. |
|---|---|---|---|---|---|---|---|---|
| | - | Plus | - | Plus | | Full | Hard | |
| **AR** | | | | | | | | |
| Qwen2.5-Coder | 61.6 | 51.8 | 75.9 | 61.4 | 56.6 | **46.1** | 16.2 | 52.2 |
| OpenCoder (Huang et al., 2024) | 66.5 | **63.4** | **79.9** | **70.4** | 66.9 | 40.5 | 9.5 | **55.0** |
| **Diffusion** | | | | | | | | |
| LLaDA (Nie et al., 2025) | 35.4 | 30.5 | 50.1 | 42.1 | 36.3 | 18.9 | 4.1 | 30.2 |
| Dream (Ye et al., 2025) | 56.7 | 50.0 | 68.7 | 57.4 | 53.7 | 23.6 | 4.1 | 43.4 |
| DiffuCoder | 67.1 | 60.4 | 74.2 | 60.9 | 60.7 | 40.2 | 12.8 | 52.6 |
| CADD (ours) | 72.0 | 63.4 | **75.7** | **63.2** | **63.3** | **42.1** | **17.6** | **55.7** |
| CADD (ours, DiffuCoder init) | **73.8** | **64.6** | 73.9 | 60.4 | 62.5 | 41.5 | 15.5 | 55.0 |

10.04 with 512 function evaluations (NFE), CADD surpasses the MDM variants by 0.38 in terms of FID and represents the best result among all compared method. On ImageNet-32, as shown in Table 2, the observation is constent, where CADD obtains FID of 3.74 and outperforms all reported baselines. The qualitative generated samples are provided in Appendix D for visual justifications.

## 5.3 CODE GENERATION

For a large-scale setting, we conduct code generation experiments based on the DiffuCoder pipeline (Gong et al., 2025b). The DiffuCoder base model training process involves adapting a pretrained autoregressive LLM (e.g., Qwen2.5-coder (Hui et al., 2024)) into a discrete diffusion model by annealing its attention mechanism from causal to bidirectional (Gong et al., 2025a). The resulting model is then trained using a masking diffusion loss (Shi et al., 2024). In this context, we evaluate our method using the following two distinct configurations. (i) Vanilla CADD: We follow the DiffuCoder procedure to adapt the Qwen2.5-coder model. Instead of using the MDM loss, we train the model from the beginning with our proposed CADD loss. (ii) CADD (fine-tuned): To demonstrate CADD's effectiveness as a fine-tuning objective, we initialize our model from a pretrained DiffuCoder checkpoint and then continue training it with the CADD loss. To ensure a fair comparison, both CADD variants are trained on the same 65B total tokens and use the same training hyperparameters as the original DiffuCoder. In the evaluation, we follow their settings to test the model performance on three coding benchmarks: HumanEval (Chen et al., 2021), MBPP (Austin et al., 2021b), and BigCodeBench (Zhuo et al., 2024).

Table 3 reports the pass@1 performance, where the results of both autoregressive (AR) and diffusion-based LLMs are included, with an overall average score provided. Compared with Diffusion-based models, CADD emerges as the strongest diffusion model, outperforming competitors on nearly all metrics. Compared to the previous leading DM, DiffuCoder, CADD significantly improves performance on HumanEval, e.g., from 67.1 to 72.0; on the challenging BigCodeBench-Hard subset, we can also observe significant performance gain from 12.8 to 17.6. CADD is also highly competitive with leading AR code models. It surpasses Qwen2.5-Coder across all benchmarks and achieves a higher overall average than OpenCoder (55.7 vs. 55.0).

## 5.4 ABLATION STUDIES

**On the choice of fusion and $\hat{z}_0$ estimation**  In most of our experiments, we choose to fuse the discrete mask token embedding and continuous embedding with addition operation, i.e., $\tilde{z}_t = z_{\text{disc}} + z_t$. We consider two extra manners to fuse these two domains: 1) concatenation $[z_{\text{disc}}, z_t]$; 2) reweighted sum $\alpha_t z_{\text{disc}} + (1 - \alpha_t) z_t$, where $\alpha_t$ decreases as the position is more likely to be clean (unmasked). The intuition is that when a token is unlikely to be masked, the model should lean more on $z_t$ to carry semantic content, hence a smaller $\alpha_t$.

Observing the results in Table 4, MAUVE varies by only 0.03 absolute and Entropy varies by 0.07 absolute across the different choices. These three options do not show significant difference to the

Table 4: Performance vs. fusion method for $\tilde{z}_t$

| Fusion | MAUVE ($\uparrow$) | Entropy ($\uparrow$) |
|---|---|---|
| Add | 0.24 | 5.31 |
| Concate | 0.21 | 5.37 |
| Reweight | 0.24 | 5.30 |

Table 5: Performance vs. continuous schedules

| Metrics | FM | VP |
|---|---|---|
| MAUVE | 0.24 | 0.24 |
| FiD | 2.88 | 3.16 |

Table 6: Performance vs. training and estimation method for $\hat{z}_0$

| Metric | CE | CE + MSE | |
|---|---|---|---|
| | | hard $\hat{z}_0$ | soft $\hat{z}_0$ |
| MAUVE | | | |
| hard $\hat{z}_0$ | 0.24 | 0.24 | 0.22 |
| soft $\hat{z}_0$ | 0.18 | 0.24 | 0.24 |
| computation cost | | | |
| TFLOPS | 0.291 | 0.325 | 0.291 |
| TPS/GPU | 47,152 | 32,117 | 47,152 |

performance, while concatenation involves an additional projection layer to match the embedding dimension.

Morever, we compare the choice of $\hat{z}_0$ estimation, as discussed in Eq. (21):

$$\textbf{hard: } \hat{x}_0 = \arg\max_v \pi_{\theta,i}(v), \ \hat{z}_0 = w_\theta(x_0) \quad \textbf{soft: } \hat{z}_{0,\theta} := \sum_v p_\theta(\hat{x}_0 = v \mid x_t, z_t) \, w_{\theta,v}.$$

From Table 6, hard estimation gives higher MAUVE (+0.06) and slightly lower Entropy (-0.11), indicating this choice is mode-seeking-oriented, where the context is localized faster. The soft estimation encounter shows higher entropy, meaning that the model reveals a mode-covering behavior and it pursue a better diversity for generation. The properties of these choices are justified. We consider both options are valid for the sampling, depending on which properties we are looking into in practical case.

**On the choice of training objective** To further justify whether we should use MSE to optimize at the embedding level in the categorical generative modeling scenario. We add training results that include an extra MSE loss $\|\hat{z}_0 - z_0\|^2$ using the two parameterizations of $\hat{z}_0$ (soft and hard) defined in Eq. (21). From the results, CE + MSE provides performance that is close to using CE alone. The soft parameterization gives a gain under hard-inference MAUVE, but it introduces higher computation cost. This is because the soft prediction requires a matrix multiplication between the predicted probability vector $\mathbb{R}^{B \times L \times d}$ and the token embedding matrix $\mathbb{R}^{d \times V}$ with batch size $B$, sequence length $L$, embedding dimension $d$ and vocabulary size $V$. Such extra cost reduces TPS/GPU and increases TFLOPS. A possible explanation for the limited improvement is that, in the categorical setting, an MSE loss behaves similarly to cross entropy since both losses guide the model toward selecting the correct token and its embedding. We expect MSE to be more useful in settings where the targets are not purely categorical.

**On the choice of continuous schedule** We compare the Variance-Preserving (VP) schedule with Flow-matching (FM) schedule in both text and image generation experiments. The results are shown in Table 5. For text generation, the results are on par with the Flow-matching schedule. For image generation, the FiD score is slightly worse under the VP schedule. This difference may come from the need for different hyper-parameter settings for the two schedules.

## 6 CONCLUSION

In standard discrete diffusion, information is lost abruptly when tokens are replaced by an absorbing state. Inspired by Gaussian diffusion, where the data signal degrades smoothly, CADD's core idea is to introduce an auxiliary continuous space to guide the discrete process. This space is designed to retain semantic information, providing a smooth continuous representation of a token even after its discrete form has been absorbed. By conditioning on it, the model can better "remember" what was supposed to be in the masked position. This leads to more coherent and contextually accurate generations, as the model has a stronger grasp of the underlying meaning. With extensive empirical justification on text, image and code generation, we justify that with the continuous augmented space proposed in CADD, the discrete diffusion models consistently generate higher quality samples across these different tasks and achieve strong performance.

ACKNOWLEDGMENT

The authors thank Josh Susskind, Irina Belousova, Miguel Angel Bautista, Richard Bai, Shuangfei Zhai, Tatiana Likhomanenko, Xiaoming Zhao, Yuyang Wang and Zijin Gu for insightful feedbacks and discussions.

ETHICS STATEMENT

This research proposes Continuously Augmented Discrete Diffusion model, for discrete data generation, with possible impacts on misusage to generate toxic information. Since our research scope is on the fundamental machine learning algorithm, we use all public datasets, which do not contain personal and sensitive information. In our paper, all generated results have been checked and do not contain misleading and malicious information.

REPRODUCIBILITY STATEMENT

The authors are committed to the principle of reproducibility and have made every effort to ensure our theoretical and experimental results can be reproduced. We confirm the variables used for the derivations are well defined and the claims are provided with proofs, which are attached in Appendix A. We have introduced our methods with precise math tools and visual aids, such as Figure 1, Figure 2, and Algorithm box 1, 2. The datasets we used are all public and widely-used. The training and inference details are described in Appendix B, including data pre-processing, training hyper-parameters, and inference pipeline. The source-code of our model will be released upon acceptance.

THE USE OF LARGE LANGUAGE MODELS (LLMS)

We acknowledge that LLMs are used to only polish the presentation and writing of the paper. The generated sentences are double checked and rephrased by the authors.

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

## A  DETAILED DERIVATIONS AND PROOF

### A.1  ELBO DERIVATION

**Forward chain.** For any observation $\boldsymbol{x}_0$, the forward diffusion constructs as

$$q(\boldsymbol{x}_{1:T}, \boldsymbol{z}_{1:T} \mid \boldsymbol{x}_0) = \prod_{t=1}^{T} q_t(\boldsymbol{x}_t, \boldsymbol{z}_t \mid \boldsymbol{x}_{t-1}, \boldsymbol{z}_{t-1}, \boldsymbol{x}_0), \tag{22}$$

note we represent $(\boldsymbol{x}_0, \boldsymbol{z}_0)$ as $\boldsymbol{x}_0$ since the transform $\boldsymbol{w}_\theta$ is deterministic.

**Reverse generative model.**

$$p_\theta(\boldsymbol{x}_0, \boldsymbol{x}_{1:T}, \boldsymbol{z}_{1:T}) = p_T(\boldsymbol{x}_T, \boldsymbol{z}_T) \Big[ \prod_{t=2}^{T} p_\theta(\boldsymbol{x}_{t-1}, \boldsymbol{z}_{t-1} \mid \boldsymbol{x}_t, \boldsymbol{z}_t) \Big] p_\theta(\boldsymbol{x}_0 \mid \boldsymbol{x}_1, \boldsymbol{z}_1). \tag{23}$$

**Proposition 3** (ELBO decomposition). *Given the forward chain $q$ defined in Eq. (22) and reverse model $p_\theta$ in Eq. (23), we have the decomposed ELBO as following:*

$$\log p_\theta(\boldsymbol{x}_0) \geq \underbrace{\mathbb{E}_{q(\boldsymbol{x}_1, \boldsymbol{z}_1 \mid \boldsymbol{x}_0)}\big[ \log p_\theta(\boldsymbol{x}_0 \mid \boldsymbol{x}_1, \boldsymbol{z}_1) \big]}_{\text{reconstruction term at } t=1}$$

$$\underbrace{- \sum_{t=2}^{T} \mathbb{E}_{q(\boldsymbol{x}_t, \boldsymbol{z}_t \mid \boldsymbol{x}_0)}\Big[ D_{\mathrm{KL}}\big( q(\boldsymbol{x}_{t-1}, \boldsymbol{z}_{t-1} \mid \boldsymbol{x}_t, \boldsymbol{z}_t, \boldsymbol{x}_0) \,\|\, p_\theta(\boldsymbol{x}_{t-1}, \boldsymbol{z}_{t-1} \mid \boldsymbol{x}_t, \boldsymbol{z}_t)) \big) \Big]}_{\text{denoising matches for } t>1}$$

$$\underbrace{- D_{\mathrm{KL}}\big( q(\boldsymbol{x}_T, \boldsymbol{z}_T \mid \boldsymbol{x}_0) \,\|\, p_T(\boldsymbol{x}_T, \boldsymbol{z}_T) \big)}_{\text{prior match at } T}. \tag{24}$$

*If $q(\boldsymbol{x}_T, \boldsymbol{z}_T \mid \boldsymbol{x}_0) = p_T(\boldsymbol{x}_T, \boldsymbol{z}_T)$ for all $\boldsymbol{x}_0$, then the prior match term is zero. The bound is tight if and only if*

$$p_\theta(\boldsymbol{x}_{t-1}, \boldsymbol{z}_{t-1} \mid \boldsymbol{x}_t, \boldsymbol{z}_t) = q(\boldsymbol{x}_{t-1}, \boldsymbol{z}_{t-1} \mid \boldsymbol{x}_t, \boldsymbol{z}_t, \boldsymbol{x}_0) \quad \text{for all } t \geq 2,$$

*and the prior match is zero, and the decoder $p_\theta(\boldsymbol{x}_0 \mid \boldsymbol{x}_1, \boldsymbol{z}_1)$ equals the true conditional induced by the joint.*

Recap the forward kernel defined in Eq. (7) and Eq. (8):

$$q(\boldsymbol{x}_t \mid \boldsymbol{x}_{t-1}) = \prod_{i=1}^{n} \mathrm{Categorical}\big( \boldsymbol{x}_t^i; \boldsymbol{Q}_t^\top \boldsymbol{x}_{t-1}^i \big), \quad \boldsymbol{Q}_t = (1 - \beta_t)\boldsymbol{I} + \beta_t \, \mathbf{1} \, \boldsymbol{m}^\top.$$

$$q(\boldsymbol{z}_t \mid \boldsymbol{z}_{t-1}, \boldsymbol{x}_{t-1}, \boldsymbol{x}_t, \boldsymbol{x}_0) = \prod_{i=1}^{n} \begin{cases} \delta(\boldsymbol{z}_t^i - \boldsymbol{z}_{t-1}^i), & \boldsymbol{x}_t^i \neq \boldsymbol{m}, \\ \mathcal{N}\big( \boldsymbol{z}_t^i; \ \sqrt{\bar{\gamma}_t} \, \boldsymbol{z}_{t-1}^i, \ (1 - \bar{\gamma}_t)\boldsymbol{I}_d \big), & \boldsymbol{x}_t^i = \boldsymbol{m}, \boldsymbol{x}_{t-1}^i \neq \boldsymbol{m}, \\ \mathcal{N}\big( \boldsymbol{z}_t^i; \ \sqrt{\gamma_t} \, \boldsymbol{z}_{t-1}^i, \ (1 - \gamma_t)\boldsymbol{I}_d \big), & \boldsymbol{x}_t^i = \boldsymbol{m}, \boldsymbol{x}_{t-1}^i = \boldsymbol{m}. \end{cases}$$

*Proof of Proposition 3.* The proof is mostly done in Sohl-Dickstein et al. (2015) and Ho et al. (2020). We include the following proof to show the generalized version with added variables. Start from the evidence identity and apply Jensen inequality:

$$\log p_\theta(\boldsymbol{x}_0) = \log \int q(\boldsymbol{x}_{1:T}, \boldsymbol{z}_{1:T} \mid \boldsymbol{x}_0) \frac{p_\theta(\boldsymbol{x}_0, \boldsymbol{x}_{1:T}, \boldsymbol{z}_{1:T})}{q(\boldsymbol{x}_{1:T}, \boldsymbol{z}_{1:T} \mid \boldsymbol{x}_0)} \, d\boldsymbol{x}_{1:T} \, d\boldsymbol{z}_{1:T}$$

$$\geq \mathbb{E}_{q(\boldsymbol{x}_{1:T}, \boldsymbol{z}_{1:T} \mid \boldsymbol{x}_0)}\Big[ \log p_\theta(\boldsymbol{x}_0, \boldsymbol{x}_{1:T}, \boldsymbol{z}_{1:T}) - \log q(\boldsymbol{x}_{1:T}, \boldsymbol{z}_{1:T} \mid \boldsymbol{x}_0) \Big]$$

$$=: \mathcal{L}(\theta; \boldsymbol{x}_0). \tag{25}$$

Insert the model and forward factorizations Eq. (23) and Eq. (22):

$$\mathcal{L}(\theta; \boldsymbol{x}_0) = \mathbb{E}_q\Big[ \log p_T(\boldsymbol{x}_T, \boldsymbol{z}_T) + \sum_{t=2}^{T} \log p_\theta(\boldsymbol{x}_{t-1}, \boldsymbol{z}_{t-1} \mid \boldsymbol{x}_t, \boldsymbol{z}_t) \tag{26}$$

$$+ \log p_\theta(\boldsymbol{x}_0 \mid \boldsymbol{x}_1, \boldsymbol{z}_1) - \sum_{t=1}^{T} \log q(\boldsymbol{x}_t, \boldsymbol{z}_t \mid \boldsymbol{x}_{t-1}, \boldsymbol{z}_{t-1}, \boldsymbol{x}_0) \Big]. \tag{27}$$

For each $t \geq 2$ use Bayes' rule under $q$:

$$q(\boldsymbol{x}_t, \boldsymbol{z}_t \mid \boldsymbol{x}_{t-1}, \boldsymbol{z}_{t-1}, \boldsymbol{x}_0) = \frac{q(\boldsymbol{x}_{t-1}, \boldsymbol{z}_{t-1} \mid \boldsymbol{x}_t, \boldsymbol{z}_t, \boldsymbol{x}_0)\, q(\boldsymbol{x}_t, \boldsymbol{z}_t \mid \boldsymbol{x}_0)}{q(\boldsymbol{x}_{t-1}, \boldsymbol{z}_{t-1} \mid \boldsymbol{x}_0)}. \tag{28}$$

Taking $\mathbb{E}_q[\log(\cdot)]$ of Eq. (28) and rearranging gives, for $t \geq 2$,

$$\mathbb{E}_q\Big[ \log p_\theta(\boldsymbol{x}_{t-1}, \boldsymbol{z}_{t-1} \mid \boldsymbol{x}_t, \boldsymbol{z}_t) - \log q(\boldsymbol{x}_t, \boldsymbol{z}_t \mid \boldsymbol{x}_{t-1}, \boldsymbol{z}_{t-1}, \boldsymbol{x}_0) \Big]$$
$$= -\mathbb{E}_{q(\boldsymbol{x}_t, \boldsymbol{z}_t \mid \boldsymbol{x}_0)}\Big[ D_{\mathrm{KL}}\big(q(\boldsymbol{x}_{t-1}, \boldsymbol{z}_{t-1} \mid \boldsymbol{x}_t, \boldsymbol{z}_t, \boldsymbol{x}_0) \,\|\, p_\theta(\boldsymbol{x}_{t-1}, \boldsymbol{z}_{t-1} \mid \boldsymbol{x}_t, \boldsymbol{z}_t)) \Big]$$
$$- \mathbb{E}_q\big[ \log q(\boldsymbol{x}_t, \boldsymbol{z}_t \mid \boldsymbol{x}_0) \big] + \mathbb{E}_q\big[ \log q(\boldsymbol{x}_{t-1}, \boldsymbol{z}_{t-1} \mid \boldsymbol{x}_0) \big]. \tag{29}$$

Sum Eq. (29) over $t = 2, \ldots, T$. The last two expectations telescope:

$$-\sum_{t=2}^{T} \mathbb{E}_q\big[ \log q(\boldsymbol{x}_t, \boldsymbol{z}_t \mid \boldsymbol{x}_0) \big] + \sum_{t=2}^{T} \mathbb{E}_q\big[ \log q(\boldsymbol{x}_{t-1}, \boldsymbol{z}_{t-1} \mid \boldsymbol{x}_0) \big]$$
$$= \mathbb{E}_q\big[ \log q(\boldsymbol{x}_1, \boldsymbol{z}_1 \mid \boldsymbol{x}_0) \big] - \mathbb{E}_q\big[ \log q(\boldsymbol{x}_T, \boldsymbol{z}_T \mid \boldsymbol{x}_0) \big]. \tag{30}$$

Plug this back into Eq. (27) and group the boundary terms with $\log p_T$:

$$\mathcal{L}(\theta; \boldsymbol{x}_0) = \mathbb{E}_q\big[ \log p_\theta(\boldsymbol{x}_0 \mid \boldsymbol{x}_1, \boldsymbol{z}_1) \big]$$
$$- \sum_{t=2}^{T} \mathbb{E}_{q(\boldsymbol{x}_t, \boldsymbol{z}_t \mid \boldsymbol{x}_0)}\Big[ D_{\mathrm{KL}}\big(q(\boldsymbol{x}_{t-1}, \boldsymbol{z}_{t-1} \mid \boldsymbol{x}_t, \boldsymbol{z}_t, \boldsymbol{x}_0) \,\|\, p_\theta(\boldsymbol{x}_{t-1}, \boldsymbol{z}_{t-1} \mid \boldsymbol{x}_t, \boldsymbol{z}_t)) \Big]$$
$$- \Big( \mathbb{E}_q\big[ \log q(\boldsymbol{x}_T, \boldsymbol{z}_T \mid \boldsymbol{x}_0) \big] - \mathbb{E}_q\big[ \log p_T(\boldsymbol{x}_T, \boldsymbol{z}_T) \big] \Big)$$
$$- \mathbb{E}_q\big[ \log q(\boldsymbol{x}_1, \boldsymbol{z}_1 \mid \boldsymbol{x}_0) \big]. \tag{31}$$

Now we recoginize the prior KL to obtain

$$\mathcal{L}(\theta; \boldsymbol{x}_0) = \mathbb{E}_q\big[ \log p_\theta(\boldsymbol{x}_0 \mid \boldsymbol{x}_1, \boldsymbol{z}_1) \big]$$
$$- \sum_{t=2}^{T} \mathbb{E}_{q(\boldsymbol{x}_t, \boldsymbol{z}_t \mid \boldsymbol{x}_0)}\Big[ D_{\mathrm{KL}}\big(q(\boldsymbol{x}_{t-1}, \boldsymbol{z}_{t-1} \mid \boldsymbol{x}_t, \boldsymbol{z}_t, \boldsymbol{x}_0) \,\|\, p_\theta(\boldsymbol{x}_{t-1}, \boldsymbol{z}_{t-1} \mid \boldsymbol{x}_t, \boldsymbol{z}_t)) \Big]$$
$$- D_{\mathrm{KL}}\big(q(\boldsymbol{x}_T, \boldsymbol{z}_T \mid \boldsymbol{x}_0) \,\|\, p_T(\boldsymbol{x}_T, \boldsymbol{z}_T)\big) \; - \; \underbrace{\mathbb{E}_q\big[ \log q(\boldsymbol{x}_1, \boldsymbol{z}_1 \mid \boldsymbol{x}_0) \big]}_{=:C(\boldsymbol{x}_0)}. \tag{32}$$

Note the last term $C(\boldsymbol{x}_0)$ does not involve $p_\theta$ and can be dropped, and we normally do not optimize the last KL term $D_{\mathrm{KL}}\big(q(\boldsymbol{x}_T, \boldsymbol{z}_T \mid \boldsymbol{x}_0) \,\|\, p_T(\boldsymbol{x}_T, \boldsymbol{z}_T)\big)$ as we let the schedule to make this statistical distance is sufficiently small. □

## A.2 FORWARD

We can derive the following lemma for the marginal at time step $t$.

**Lemma 2** (Continuous marginal conditioned on $(\boldsymbol{x}_t, \boldsymbol{x}_0)$). *Let $\bar{\gamma}_t := \prod_{s=1}^{t} \gamma_s$. For each position $i$, we have continuous marginal conditioned on $(\boldsymbol{x}_t, \boldsymbol{x}_0)$ as*

$$q(\boldsymbol{z}_t^i \mid \boldsymbol{x}_t^i, \boldsymbol{x}_0^i) = \begin{cases} \delta(\boldsymbol{z}_t^i - \boldsymbol{z}_0^i), & \boldsymbol{x}_t^i = \boldsymbol{x}_0^i, \\ \mathcal{N}(\boldsymbol{z}_t^i;\, \sqrt{\bar{\gamma}_t}\, \boldsymbol{z}_0^i,\, (1 - \bar{\gamma}_t)\boldsymbol{I}_d), & \boldsymbol{x}_t^i = \boldsymbol{m}, \end{cases}$$

*with $\boldsymbol{z}_0^i = \boldsymbol{w}_\theta(\boldsymbol{x}_0^i)$. Hence We finally have*

$$q(\boldsymbol{z}_t \mid \boldsymbol{x}_t, \boldsymbol{x}_0) = \prod_{i=1}^{n} q(\boldsymbol{z}_t^i \mid \boldsymbol{x}_t^i, \boldsymbol{x}_0^i) = \Big[ \prod_{i:\, \boldsymbol{x}_t^i \neq \boldsymbol{m}} \delta(\boldsymbol{z}_t^i - \boldsymbol{z}_0^i) \Big] \cdot \Big[ \prod_{i:\, \boldsymbol{x}_t^i = \boldsymbol{m}} \mathcal{N}(\boldsymbol{z}_t^i;\, \sqrt{\bar{\gamma}_t}\boldsymbol{z}_0^i, (1 - \bar{\gamma}_t)\boldsymbol{I}_d) \Big].$$

Then what follows proves Proposition 2. We first prove the conditional independency between $\boldsymbol{z}_t$ and $\boldsymbol{x}_{t-1}$ given $(\boldsymbol{x}_t, \boldsymbol{x}_0)$ in the reverse context.

**Lemma 3** (Conditional independency between $\boldsymbol{z}_t$ and $\boldsymbol{x}_{t-1}$ given $(\boldsymbol{x}_t, \boldsymbol{x}_0)$)**.** $\boldsymbol{z}_t$ and $\boldsymbol{x}_{t-1}$ are conditionally independent given $(\boldsymbol{x}_t, \boldsymbol{x}_0)$ based on the forward kerned defined in Eq. (8).

To prove Proposition 1, we first prove the following lemma:

*Proof of Lemma 2 and Lemma 3.* If $\boldsymbol{x}_t^i = \boldsymbol{x}_0^i$ then the absorbing chain implies $\boldsymbol{x}_s^i \neq \boldsymbol{m}$ for $s \leq t$, so the kernel gives $\boldsymbol{z}_t^i = \boldsymbol{z}_0^i$ almost surely, which is the first line of Eq. (8).

If $\boldsymbol{x}_t^i = \boldsymbol{m}$, use the law of total probability over $\boldsymbol{x}_{t-1}^i \in \{\boldsymbol{x}_0^i, \boldsymbol{m}\}$.

When $\boldsymbol{x}_{t-1}^i \neq \boldsymbol{m}$ (first time masking at $t$), the second branch of the kernel gives $\boldsymbol{z}_t^i \sim \mathcal{N}(\sqrt{\bar{\gamma}_t}\,\boldsymbol{z}_0^i, (1-\bar{\gamma}_t)\boldsymbol{I})$.

When $\boldsymbol{x}_{t-1}^i = \boldsymbol{m}$ (already masked), the third branch composes a diffusion forward step with the previous marginal $\boldsymbol{z}_{t-1}^i \sim \mathcal{N}(\sqrt{\bar{\gamma}_{t-1}}\,\boldsymbol{z}_0^i, (1-\bar{\gamma}_{t-1})\boldsymbol{I})$, which yields

$$\boldsymbol{z}_t^i \sim \mathcal{N}\big(\sqrt{\gamma_t\bar{\gamma}_{t-1}}\,\boldsymbol{z}_0^i, \ (1-\gamma_t\bar{\gamma}_{t-1})\boldsymbol{I}\big) = \mathcal{N}\big(\sqrt{\bar{\gamma}_t}\,\boldsymbol{z}_0^i, \ (1-\bar{\gamma}_t)\boldsymbol{I}\big).$$

This proves the masked line of Eq. (8). $\qquad\square$

Then leveraging these results, we can easily prove Proposition 1.

*Proof of Proposition 1.* Expand the path marginal, use Eq. (6) and Lemma 2, and factor over positions. The sum over discrete paths yields $q(\boldsymbol{x}_t \mid \boldsymbol{x}_0)$; conditioning on $\boldsymbol{x}_t$ reduces the continuous part to Lemma 2. $\qquad\square$

### A.3 REVERSE

*Proof of Proposition 2.* We first prove the factorization shown in Eq. (14). To achieve this, we just need to show:

$$q(\boldsymbol{x}_{t-1}, \boldsymbol{z}_{t-1} \mid \boldsymbol{x}_t, \boldsymbol{z}_t, \boldsymbol{x}_0) = q(\boldsymbol{x}_{t-1} \mid \boldsymbol{x}_t, \boldsymbol{z}_t, \boldsymbol{x}_0) \cdot q(\boldsymbol{z}_{t-1} \mid \boldsymbol{x}_t, \boldsymbol{z}_t, \boldsymbol{x}_{t-1}, \boldsymbol{x}_0) \tag{33}$$

$$= \frac{q(\boldsymbol{z}_t \mid \boldsymbol{x}_{t-1}, \boldsymbol{x}_t, \boldsymbol{x}_0)q(\boldsymbol{x}_{t-1} \mid \boldsymbol{x}_t, \boldsymbol{x}_0)}{q(\boldsymbol{z}_t \mid \boldsymbol{x}_t, \boldsymbol{x}_0)} \cdot q(\boldsymbol{z}_{t-1} \mid \boldsymbol{x}_t, \boldsymbol{z}_t, \boldsymbol{x}_{t-1}, \boldsymbol{x}_0) \tag{34}$$

$$= q(\boldsymbol{x}_{t-1} \mid \boldsymbol{x}_t, \boldsymbol{x}_0) \cdot q(\boldsymbol{z}_{t-1} \mid \boldsymbol{x}_t, \boldsymbol{z}_t, \boldsymbol{x}_{t-1}, \boldsymbol{x}_0), \tag{35}$$

where $q(\boldsymbol{z}_t \mid \boldsymbol{x}_{t-1}, \boldsymbol{x}_t, \boldsymbol{x}_0) = q(\boldsymbol{z}_t \mid \boldsymbol{x}_t, \boldsymbol{x}_0)$ by the conditional independence according to Lemma 3. Then the discrete part is the same as discrete diffusion, we may leverage the results from Austin et al. (2021a); Sahoo et al. (2024); Shi et al. (2024) to complete the proof of Eq. (15).

Next, we prove the closed form of the continuous part, $q(\boldsymbol{z}_{t-1} \mid \boldsymbol{x}_t, \mathbf{z}_t, \boldsymbol{x}_{t-1}, \boldsymbol{x}_0)$, by case analysis based on the discrete states. We start with Bayes' rule for the continuous variables:

$$q(\boldsymbol{z}_{t-1} \mid \boldsymbol{x}_t, \boldsymbol{z}_t, \boldsymbol{x}_{t-1}, \boldsymbol{x}_0) \propto q(\boldsymbol{z}_t \mid \boldsymbol{z}_{t-1}, \boldsymbol{x}_t) \cdot q(\boldsymbol{z}_{t-1} \mid \boldsymbol{x}_{t-1}, \boldsymbol{x}_0). \tag{36}$$

The forms of the two terms on the right-hand side are Gaussian distributions, but will change depending on the discrete states and it leads to the three cases.

**Case 1: No mask at $t$ ($\boldsymbol{x}_t = \boldsymbol{x}_0$).** In this case, no noise has been applied to the embedding up to timestep t-1. Thus, both terms directly have a Dirac delta function: $q(\boldsymbol{z}_{t-1} \mid \boldsymbol{x}_{t-1} = \boldsymbol{x}_0, \boldsymbol{x}_0) = \delta(\boldsymbol{z}_{t-1} - \boldsymbol{z}_0)$. The posterior is therefore also a Dirac delta function, proving the first part of Eq. (16).

**Case 2: First time unmask at $t$ ($\boldsymbol{x}_t = \boldsymbol{m}$, $\boldsymbol{x}_{t-1} = \boldsymbol{x}_0$).** In this case, the first term in Eq. (36) is Gaussian while the second term becomes a Dirac $\delta(\boldsymbol{z}_{t-1} - \boldsymbol{z}_0)$. The multiplication yields a Dirac posterior at the same point: $q(\boldsymbol{z}_{t-1} \mid \boldsymbol{x}_{t-1} = \boldsymbol{x}_0, \boldsymbol{x}_0) = \delta(\boldsymbol{z}_{t-1} - \boldsymbol{z}_0)$.

**Case 3: Remaining masked at $t$ ($\boldsymbol{x}_t = \boldsymbol{m}$, $\boldsymbol{x}_{t-1} = \boldsymbol{m}$).** In this case, both terms remain in Gaussian distribution, and the parameters are same with normal Gaussian diffusion models. The product of these two Gaussians is a new Gaussian, allowing usu to use the standard derivation for DDPM (Ho et al., 2020), by completing the square on the exponent, we find that the resulting distribution is $\mathcal{N}(\mathbf{z}_{t-1}; \tilde{\boldsymbol{\mu}}_t(\mathbf{z}_t, \mathbf{z}_0), \ \tilde{\beta}_t\mathbf{I})$, which proves the last part of Eq. (16).

$\qquad\square$

*Proof of Lemma 1.* Using the results from Proposition 2, for a single position $i$, the exact one–step KL at timestep $t > 1$ inside the ELBO is

$$D_{\text{KL}}(\boldsymbol{x}_0, t) := \mathbb{E}_{q(\boldsymbol{x}_t, \boldsymbol{z}_t|\boldsymbol{x}_0)}\Big[D_{\text{KL}}\big(q(\boldsymbol{x}_{t-1}, \boldsymbol{z}_{t-1} \mid \boldsymbol{x}_t, \boldsymbol{z}_t, \boldsymbol{x}_0) \,\big\|\, p_\theta(\boldsymbol{x}_{t-1}, \boldsymbol{z}_{t-1} \mid \boldsymbol{x}_t, \boldsymbol{z}_t))\Big], \quad (37)$$

For the unmasked positions ($\boldsymbol{x}_t \neq \boldsymbol{m}$), the KL is identically 0, and plug in Eq. (14), 15 and 16, we recover Eq. (18) exactly as

$$D_{\text{KL}}\big(q(\cdot \mid \boldsymbol{x}_t, \boldsymbol{z}_t, \boldsymbol{x}_0) \,\big\|\, p_\theta(\cdot \mid \boldsymbol{x}_t, \boldsymbol{z}_t)\big) = \underbrace{\rho_t^{\text{flip}}\big[-\log p_\theta(\boldsymbol{x}_0|\boldsymbol{x}_t, \boldsymbol{z}_t)\big]}_{\text{discrete}} + \underbrace{\rho_t^{\text{keep}}\, \mathcal{D}_{\text{KL}}{}^{\text{cont}}}_{\text{continuous}},$$

with the ratio that determines whether the position is going to be flipped to unmask:

$$\rho_t^{\text{keep}} = \frac{1 - \alpha_{t-1}}{1 - \alpha_t}, \qquad \rho_t^{\text{flip}} = \frac{\alpha_{t-1}\beta_t}{1 - \alpha_t} = \frac{\alpha_{t-1} - \alpha_t}{1 - \alpha_t}. \qquad (38)$$

The discrete KL part exactly recovers the results from the absorbing discrete diffusion models (Austin et al., 2021a; Sahoo et al., 2024; Shi et al., 2024), and the continuous KL divergence:

$$D_{\text{KL}}^{\text{cont}} = D_{\text{KL}}\Big(\mathcal{N}(\boldsymbol{\mu}^\star, \tilde{\beta}_t \boldsymbol{I}_d) \,\big\|\, \mathcal{N}(\boldsymbol{\mu}_v, \tilde{\beta}_t \boldsymbol{I}_d)\Big), \quad \boldsymbol{\mu}^\star = \tilde{\boldsymbol{\mu}}_t(\boldsymbol{z}_0, \boldsymbol{z}_t), \;\; \boldsymbol{\mu}_v = \tilde{\boldsymbol{\mu}}_t(\hat{\boldsymbol{z}}_0, \boldsymbol{z}_t), \qquad (39)$$

where we recap

$$\tilde{\boldsymbol{\mu}}_t(\zeta, \boldsymbol{z}_t) = \frac{\sqrt{\bar{\gamma}_{t-1}}(1 - \gamma_t)}{1 - \bar{\gamma}_t}\zeta + \frac{\sqrt{\gamma_t}(1 - \bar{\gamma}_{t-1})}{1 - \bar{\gamma}_t}\boldsymbol{z}_t, \qquad \tilde{\beta}_t = \frac{(1 - \bar{\gamma}_{t-1})(1 - \gamma_t)}{1 - \bar{\gamma}_t}.$$

This results in the comparison between $\boldsymbol{z}_0$ and $\hat{\boldsymbol{z}}_0$ and the KL divergence reduced to:

$$D_{\text{KL}}^{\text{cont}} = \frac{1}{2\tilde{\beta}_t}\left\|\tilde{\boldsymbol{\mu}}_t(\boldsymbol{z}_0, \boldsymbol{z}_t) - \tilde{\boldsymbol{\mu}}_t(\hat{\boldsymbol{z}}_{0,\theta}, \boldsymbol{z}_t^i)\right\|^2 = \frac{a_t^2}{2\tilde{\beta}_t}\left\|\boldsymbol{z}_0 - \hat{\boldsymbol{z}}_{0,\theta}\right\|^2; \; a_t = \frac{\sqrt{\bar{\gamma}_{t-1}}(1 - \gamma_t)}{1 - \bar{\gamma}_t}.$$

$\square$

**Remark 1** (On the Alternative Factorization). *One could also decompose the posterior using the alternative order from the chain rule:*

$$q(\boldsymbol{x}_{t-1}, \boldsymbol{z}_{t-1} \mid \cdot) = q(\boldsymbol{z}_{t-1} \mid \boldsymbol{x}_t, \boldsymbol{z}_t, \boldsymbol{x}_0) \cdot q(\boldsymbol{x}_{t-1} \mid \boldsymbol{x}_t, \boldsymbol{z}_t, \boldsymbol{z}_{t-1}, \boldsymbol{x}_0).$$

*While mathematically valid and could provide new properties in the sampling, this factorization is not fully tractable. The first term, $q(\boldsymbol{z}_{t-1}|\cdot)$, is a complex Gaussian Mixture Model. More critically, the second term, $q(\boldsymbol{x}_{t-1}|\cdot)$, has no analytical closed form, as it would require inverting the continuous diffusion process and the embedding function to infer a discrete state. The factorization in Prop. 2 is therefore adopted as a tractable choice for a more efficient algorithm implementation.*

## B   DETAILED EXPERIMENT SETTINGS

### B.1   DIFFUSION SETTINGS

The CADD forward process has two coupled components, each with its own schedule.

- Discrete schedule: we adopt the MDLM log-linear masking schedule for the discrete process (Sahoo et al., 2024). The discrete forward corruption uses a continuous-time $\alpha(t) = 1 - t$, with $t \in [0, 1]$.

- Continuous schedule: to keep the meaning of time aligned, we set the continuous latent $\boldsymbol{z}$ to follow a linear flow-matching path to isotropic noise (Lipman et al., 2022), i.e., if the position is masked, we have $\boldsymbol{z}_t = (1 - t)\boldsymbol{z}_0 + t\boldsymbol{\epsilon}$, $\boldsymbol{\epsilon} \sim \mathcal{N}(\boldsymbol{0}, \boldsymbol{I})$.

- Multi-sample estimation: we by default set $K = 1$ for the estimation of $\hat{\boldsymbol{x}}_{0,\theta}$ for fair comparison with the baselines. We provide ablation studies to demonstrate the effect of $K > 1$.

Table 7: Benchmark computation cost of discrete (mask) diffusion and CADD.

| Setting | Throughputs (Tokens/s - per GPU) | TFLOPS | TFLOPS/s | Avg. GPU Mem (GB) |
|---|---|---|---|---|
| **Training** | | | | |
| Discrete Diffusion | 47,416 | 0.29 | 6.049 | 1.362 |
| CADD | 47,152 | 0.291 | 6.082 | 1.492 |
| **Inference** | | | | |
| $k = 1$ | 47,373 | 0.291 | 6.082 | 1.492 |
| $k = 2$ | 24,852 | 0.581 | 12.206 | 3.511 |
| $k = 3$ | 15,768 | 0.873 | 18.310 | 5.253 |
| $k = 4$ | 1,417 | 1.162 | 24.417 | 7.470 |

## B.2 EXPERIMENT-SPECIFIC SETTINGS

**Text Generation** In our main experiments, including ablation studies that used to explore the properties of CADD, we train the models on OpenWebText. Following the standard MDLM pre-processing (Sahoo et al., 2024), we use the GPT-2 tokenizer, resulting in a vocabulary of 50,257 tokens. The sequence length is fixed at 1,024. Our text model is a 12-layer DiT with 12 attention heads and an embedding dimension of 768, totaling approximately 168M parameters. During training, we keep the same training configuration, i.e., we train for about 2M steps with a batch size of 256 to match the total 262B tokens seen in the training. We use the AdamW optimizer with a learning rate warmed up from 0 to $3 \times 10^{-4}$. The results in Table 8 and Table 9, are based on Text8 and LM1B dataset, where we strictly follow the training setting in Jo & Hwang (2025) and Sahoo et al. (2024). Please refer their experiment settings for more details. For evaluation, we follow ReMDM (Wang et al., 2025)'s evaluation setting, where we randomly sample 5,000 text samples with length $n = 1,024$, using $\{128, 256, 512, 1024, 4096\}$ sampling steps. The sampled token sequences are used to compute MAUVE score, generative perplexity with GPT2-Large model, and entropy.

**Image Generation** We experiment on CIFAR-10 and ImageNet (with resolution $32 \times 32$), which consists of 50,000 and 1,281,149 natural images respectively. CIFAR-10 already has $32 \times 32$ resolution. For ImageNet images, we follow the preprocessing used in EDM (Karras et al., 2022), i.e., using center-crop to make it as squared image and rescale to the desired $32 \times 32$ resolution. As the model is trained on pixel space, we treat each pixel as a discrete token, resulting in a vocabulary size 256 at each position. We follow the architecture design used in MDM-Prime (Chao et al., 2025), which is a U-Net architecture based on ADM (Dhariwal & Nichol, 2021). For CIFAR-10, we leverage an augmentation pipeline proposed in Karras et al. (2020), but only keep the rotation and flip operation to avoid pixel value changes. We let set the augmentation probability as 15% on CIFAR-10, and there is no augmentation used on ImageNet. For both experiments, we set learning rate as $1 \times 10^{-4}$ using AdamW optimizer, and train the model until it has seen 200M and 4B images respectively. In sampling, we adopt a cosine decay for temperature with $\tau_{max} = 2.5$, and applied the corrector following Gat et al. (2024). We use the standard Fréchet Inception Distance (FID) and Inception Score for evaluation, computed with 50,000 randomly generate images.

**Code Generation** We use the OpenCoder dataset (Huang et al., 2024), selected by following the recipe in DiffuCoder (Gong et al., 2025b). We strictly follow their settings to initialize the 7B model with Qwen2.5-Coder checkpoint, and adapt it to diffusion model using the techniques introduced in Gong et al. (2025a). Then we traine the model on a 64 NVIDIA A100 GPUs in total. The training process utilized BF16 mixed precision and was scaled using Fully Sharded Data Parallelism (FSDP). For optimization, we employed the Adam optimizer with a peak learning rate of $1 \times 10^{-5}$, preceded by a 2,000-step linear warmup. The model is trained with 65B tokens in total. For generation, both models were configured with a maximum sequence length of 512 tokens and a total of T=512 diffusion timesteps. During generation, we employed a top negative entropy remasking sampler. The CADD from scratch variant uses temperature 0.2 and the DiffuCoder initialized variant uses temperature 0.01.

## B.3 COMPUTATION ANALYSIS

We measured both training and inference efficiency under the same hardware (H100, FP32) and batch settings (batch size=1), using the DiT model with 169 M parameters. The results are summarized in

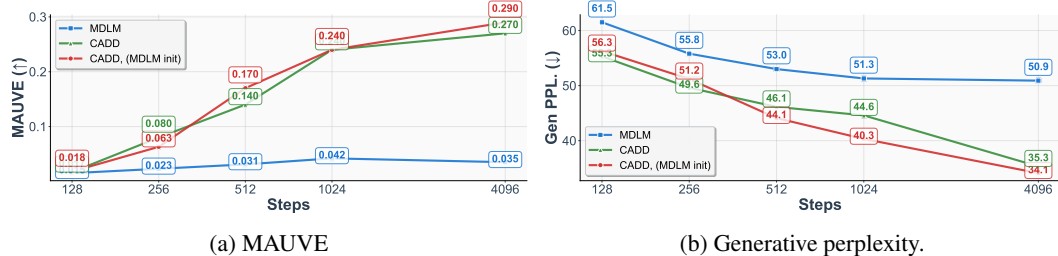

(a) MAUVE             (b) Generative perplexity.

Figure 4: Analogous figure of Figure 3. We compare the finetuned checkpoint using CADD objective with CADD and the initialization checkpoint of MDLM.

the table 7. During training, CADD matches the speed of the MDM baseline, with nearly identical token throughput (tokens/s) per gpu and memory usage. During inference, when K = 1, CADD shows similar computation cost to MDM. The inference cost increases in a linear way as K increases, since the model performs K times forward for the $z_0$ estimation, while the training cost remains unchanged.

## C    ADDITIONAL EXPERIMENT RESULTS

### C.1    TRAINING FROM MASK DIFFUSION MODEL

From the experiments on code generation, we have seen CADD could be used to finetune an existing discrete (masking) diffusion model to improve the performance. Here we provide complementary evidence that such observation is also valid on text generation. We finetune a MDLM checkpoint with CADD objective for additional 50B tokens and evaluate the performance with same setting shown in the main experiments (Figure 3). The results are shown in Figure 4. The red curve shows close performance to the green one that represent CADD's performance, which indicates CADD could efficiently finetune an existing MDM model to enhance the generation capabilities.

### C.2    PERPLEXITY EVALUATION

Since the objective of CADD involves the KL divergence of both discrete and continuous component as shown in Eq. (14), it is not fair to compare the tightness of the bound directly with other models, and we choose to focus more on the evaluation of the generated samples. However, our model is still able to compute the likelihood of the discrete part. Here we put the results for reference, aiming to provide more information to help the readers understand how the model helps the discrete diffusion side.

Table 8 and Table 9 report the perplexity evaluation on character-level and token-level respectively. The model is trained on Text8 and LM1B, following the settings of Jo & Hwang (2025) and Sahoo et al. (2024). On Text8, we can see CADD achieve very competitive perplexity results, and is slightly worse than the SoTA RDLM (Jo & Hwang, 2025). On LM1B, we can see CADD achieve the best results among diffusion models when evaluate the discrete part perplexity on both LM1B data and OWT data.

Table 10 reports the zero-shot evaluation results of the checkpoint trained on OWT data. We can observe CADD and MDLM both surpasses the perplexity of AR models on Lambada, Pubmed and Arxiv datasets. They have different dataset that they are good at in terms of perplexity, and CADD wins slightly more as it shows better zero-shot perplexity than MDLM on 4/7 tasks. These experiments result jointly indicate that CADD can not only provide strong generation quality, but also provide a good discrete likelihood bound.

### C.3    ABLATION STUDIES

**Comparing the number of samples used for $\hat{x}_0 = f_\theta(x_t, z_t^{(k)})$**    We first conduct ablation to study how the number of samples used to compute $\hat{x}_0$ would affect CADD's performance. Similar to our

Table 8: Bits Per Character (BPC) results on Text8 test set. Results are taken from Jo & Hwang (2025). Bold denotes the best result in autoregressive or diffusion models. The best diffusion results are marked in bold.

| Method | BPC ($\downarrow$) |
|---|---|
| *Autoregressive* | |
| AR | **1.23** |
| *Continuous Diffusion* | |
| Plaid | $\leq 1.48$ |
| BFN | $\leq 1.41$ |
| RDLM | $\leq$ **1.32** |
| *Discrete Diffusion* | |
| Multinomial Diffusion | $\leq 1.72$ |
| D3PM Uniform | $\leq 1.61$ |
| D3PM Absorb | $\leq 1.45$ |
| SEDD Absorb | $\leq 1.39$ |
| MDLM | $\leq 1.40$ |
| MD4 | $\leq 1.37$ |
| CADD (Ours) | $\leq 1.35$ |

Table 9: Test perplexities (PPL; $\downarrow$) on LM1B. The baseline results are taken from Sahoo et al. (2025). For CADD, we report the bound on the discrete likelihood. Best diffusion value is **bolded**. $\star$ the dataset for SEDD didn't incorporate sentence packing.

| Method | LM1B | OWT |
|---|---|---|
| *Autoregressive* | | |
| Transformer | 22.8 | 17.5 |
| *Diffusion (Uniform-state / Gaussian)* | | |
| D3PM Uniform (Austin et al., 2021a) | 137.9 | - |
| Diffusion-LM$^\star$ (Li et al., 2022) | 118.6 | - |
| SEDD Uniform (Lou et al., 2024) | 40.3$^\star$ | 29.7 |
| UDLM (Deschenaux & Gulcehre, 2025) | 36.7 | 27.4 |
| DUO (Sahoo et al., 2025) | 33.7 | 25.2 |
| *Diffusion (absorbing state)* | | |
| D3PM Absorb (Austin et al., 2021a) | 76.9 | - |
| DiffusionBert (He et al., 2023) | 63.8 | - |
| SEDD Absorb (Lou et al., 2024) | 32.7$^\star$ | 24.1 |
| MDLM (Sahoo et al., 2024) | 31.8 | 23.2 |
| CADD (Ours) | **31.4** | **23.1** |

Table 10: Zero-shot perplexities ($\downarrow$) of models trained for 1M steps on OpenWebText. All perplexities for diffusion models are upper bounds. Baseline results are taken from Sahoo et al. (2025). Best diffusion model performance results are **bolded** and diffusion values better than AR are underlined. Plaid and D3PM are trained with 0.3M more steps.

| Method | PTB | Wikitext | LM1B | Lambda | AG News | Pubmed | Arxiv |
|---|---|---|---|---|---|---|---|
| *Autoregressive* | | | | | | | |
| Transformer | 82.05 | 25.75 | 51.25 | 51.28 | 52.09 | 49.01 | 41.73 |
| *Diffusion (Uniform-state / Gaussian)* | | | | | | | |
| SEDD Unifor | 105.51 | 41.10 | 82.62 | 57.29 | 82.64 | 55.89 | 50.86 |
| Plaid | 142.60 | 50.86 | 91.12 | 57.28 | - | - | - |
| UDLM | 112.82 | 39.42 | 77.59 | 53.57 | 80.96 | 50.98 | 44.08 |
| DUO | 89.35 | 33.57 | 73.86 | 49.78 | 67.81 | 44.48 | 40.39 |
| *Diffusion (absorbing state)* | | | | | | | |
| SEDD Absorb | 100.09 | 34.28 | 68.20 | 49.86 | 62.09 | 44.53 | 38.48 |
| D3PM Absorb | 200.82 | 50.86 | 138.92 | 93.47 | - | - | - |
| MDLM | 95.26 | 32.83 | 67.01 | 47.52 | **61.15** | **41.89** | **37.37** |
| CADD (Ours) | **93.33** | **31.84** | **64.98** | **46.81** | 62.80 | 42.62 | 37.52 |

main experiments in text generation, we compare CADD with $K \in \{1, 2, 3, 4\}$ in terms of MAUVE and generative perplexity.

As shown in Figure 5, increasing both the number of sampling steps and the hyperparameter $K$ consistently improves CADD's performance. The value of $K$, which corresponds to the number of continuous samples used for soft hints, has a consistent and positive effect on generation quality. It is interesting to see the largest performance gain, especially for generative perplexity, comes from increasing $K$ from 2 to 3. The subsequent gain from $K = 3$ to $K = 4$ is smaller. One possible reason is that when $K$ is not large enough, the predicted logits could vary and make the expected value smoothed to be a flatten distribution. As $K$ gets bigger, the estimation of the correct $x_0$ becomes more accurate, resulting in better generation quality, with a trade-off between desired sample quality and inference-time latency.

We also use entropy as a complementary metric to observe the model's behavior, and the results are shown in Figure 6. We observe CADD, the highest-quality model in terms of MAUVE and generative

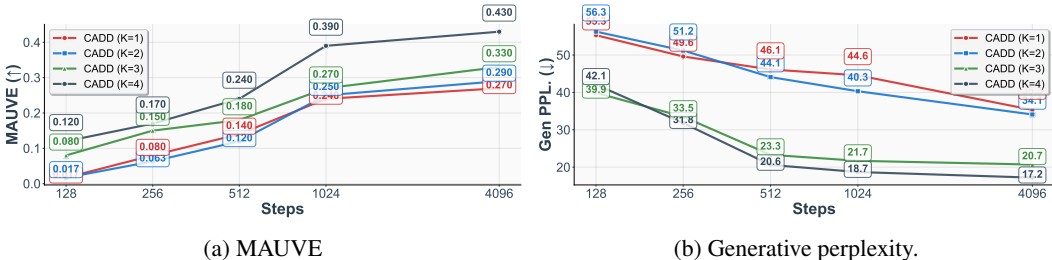

(a) MAUVE

(b) Generative perplexity.

Figure 5: Analogous figure of Figure 3. We compare of CADD variants using different number of samples to estimate $\hat{x}_0$ (K=1-4).

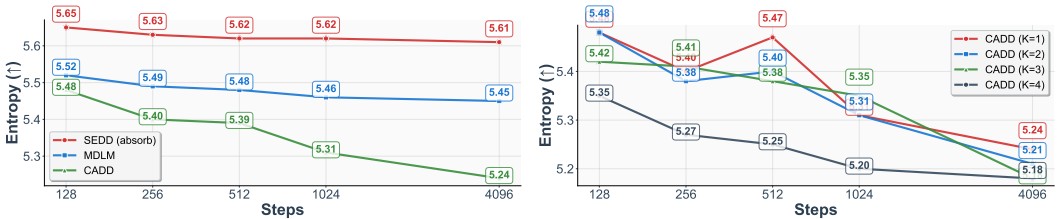

Figure 6: Analogous figure of Figure 3: study of generation variance and diversity across all methods and across different $K$. We use entropy (higher indicates more stochasticity) are reported.

perplexity (shown in Figure 3), has the lowest entropy. This indicates that CADD achieves its keeps a lower variance in the generation process with concentrating its continuous conditions. The right plot, which analyzes different values of $K$ for CADD, shows that a larger $K$ consistently leads to lower entropy. This reveals the role of $K$ as a hint mechanism. A larger $K$ provides a stronger, more deterministic "soft hint" from the continuous space, preserving smaller variance during generation. However, this does not mean CADD lack of generation diversity, as it still hits a strong MAUVE score, indicating it strikes a good balance between mode-covering and mode-seeking.

**On model architecture** Similar to the text generation, we also examine the performance of image generation. We conduct experiments to test the impacts of model architecture and number of function evaluations (NFEs) in the sampling stage. The results are reported in Table 11. As shown, ADM (Dhariwal & Nichol, 2021) shows stronger performance than DDPM++ (Song et al., 2021) across different NFEs. Especially when NFE is sufficiently large as 512, the performance of using ADM + NFE=512 configuration demonstrate a significant performance gain. As qualitative justification, we can also observe the last row of Figure 7 has the best visual quality.

|  | FID ($\downarrow$) | | |
| --- | --- | --- | --- |
| Model | 64 | 256 | 512 |
| DDPM++ | 31.24 | 4.72 | 4.70 |
| ADM | 30.41 | 4.29 | 2.88 |

Table 11: Ablation results on image generation, trained with DDPM++ and ADM architecture. FID results measured using NFE=64, 256, 512.

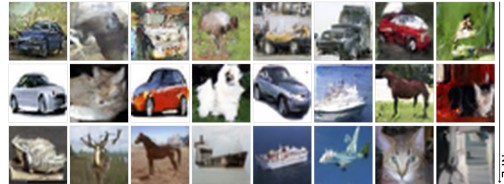

Figure 7: Qualitative results of CIFAR-10, generated by ADM, using NFE=64,256,512 (from top row to bottom).

**Qualitative visualization** Table 12 demonstrates the effect of the continuous embedding $z_t$ as a semantic scaffold that guides the discrete unmasking process. We use "A model sits on the mat" as input, and mask the second position. Using different noise level $t$ to corrupt the embedding of token "model" to form $z_t$ and predict the masked token. In the low-noise regime ($t \leq 0.3$), the continuous signal is clear enough and the prediction is same as "model". As noise increases, the embedding degrades and we can observe the predictions traverse from relevant category (e.g., vase", tank") to generic priors only when the guiding signal is fully destroyed at $t = 1.0$.

Table 12: Qualitative Generation Analysis. We visualize the prediction for the masked token in "A [MASK] sits on the mat" as we vary the noise level $t$ of the continuous embedding corrupted from the embedding "model".

| Noise Level ($t$) | Predicted Sentence | Predicted Token |
|---|---|---|
| Input (GT) | A **model** sits on the mat | **model** |
| $t = 0.1$ | A **model** sits on the mat | **model** |
| $t = 0.3$ | A **model** sits on the mat | **model** |
| $t = 0.5$ | A **vase** sits on the mat | **vase** |
| $t = 0.7$ | A **tank** sits on the mat | **tank** |
| $t = 1.0$ | A **and** sits on the mat | **and** |

## D    ADDITIONAL GENERATED SAMPLES

### D.1    TEXT GENERATION

```
As for the falling gas prices going forward, it may be similar
to declining oil prices decades ago.  Both brought low exchange
rates through decreased prices for relatively mass-produced
natural textiles such as oil and food, leading natural gas to
become extremely expensive.  With a subdued market with few
benefits being passed through marketplace costs onto consumers,
far more than was case have been spent on natural gas development,
and in order to stay on-time with rising exchange rates, state
producers had to provide cover for these higher prices.  With the
"competitive" advances inshore wind and solar--those intermittent
energy renewables, and gas --cost per kWh has fallen steadily,
driven power purchasers to switch to the cost of generating new
batteries in range efficient carbon-free natural gas.  The amount
of electricity produced per F-quarter based on natural gas dropped
by 10.4 percent and, over the 2015 Great Canada Great Average,
the gas price dropped by a one-square inch average.  Yet further
prices dropped in the subsequent years compared to 2014.  With all
the increases in wind prices compared to 2013 (when net natural
gas-fired electricity emissions fell nearly 9.5 percent)--the
average annual rate of wholesale electricity earned from the
non-resident lease-traded industries dropped from $1.262 billion
in 2015 by 37 percent, to $17.66 billion in 2015.  Residential
electricity by electric utilities increased by a $4.25 billion
more in 2015.  The U.S. is also controlled by the reduction in
the price of electricity.  Using "solar" technologies, which come
off a higher electricity prices target, have dropped by about
8 percent last year and last through to the fourth quarter, but
expect to have drastically energy prices decrease in the next
three years.While Duke Energy's nuclear project will be expected
to produce energy in the next 10 years, to the reduction in nuclear
demand that has, like, occurred in the last quarter, the nation
switched from natural gas to renewable power as their platform
for energy produce in 2015.[2] While coming to energy costs as a
whole does not in general match fossil fuel prices in large parts
of U.S., electricity prices are more competitive in the North and
North America West where price is, ever, and even where some of
these factors do so...
```

What all the results show is that they get the results that you
need.  They are showing how much more important that mindset is,
the more games you get.All MLS radio's Michael Helgen discussed
Sunday's game, noting that it put us kind of in a half time game
without help.We was not expected to finish well, so it's hard to
say that we had an opening-day performance.  We could have maybe
been surprised right away from the game that Jordan Morris (capt)
in the 76th minute changed the moment, and we ourselves sealed
a score sheet.Michael Helgen gave Rotem Winist's Swedish team
a first hand.  0:36 secondsIt felt as though the Sounders had a
special team on the team in Hawaii, and we had that sort of game
against Houston.Shelgen said a team playing at that high level as
Portland on their history earned coach Chris Porter's trust and
would take a step toward the future.Any stage of the season, for a
non-soccercer player, they're always going to find it.  It's very
rewarding to have that half of the season.  And definitely, the
MLS coach takes the game a lot, he said.  You don't need to have
a pretty good game if you're a very good player.Sometimes he has
a message that the captain may not want to do it since you've got
to ask and he's at home but rather you treat yourself more going
to practice, Helgen said.  But for us trying to leave scores open,
that was a good idea.Soccer teams said they were still impressed,
but the game affected visitors under 8.1 per cent being played off
of a Dynamo side with six clean sheets.I just think that the game
we lost against, when they started their first goal in five games,
we underestimated them, running into mistakes we could couldn't
control," Helgen said.  "But we got lucky, and I knew we'll have
to be honest to do it again, but Marko Coyle and the other smart,
athletic goalkeepers in Salford, who don't just pan through the
season, did their job and switched the bus that night.  We're a
3-0 win to Houston and a 2-1 loss.  They had to go into the locker
room, and they had to make better players, and they didn't have
space in the locker to do that, the score was 1-2."Andersen said
Denmark is preparing for the one it needs."If it's going to happen,
then we need to be confident and to do all it takes to get there
first to get the opinion, the experience, the opponents knowledge
and to make sure ourselves ready and be ready right away for the
first time," he said.  "We know that next time and practice the
way to be ready and prepared for the season."<|endoftext|>Getty
Images Brendan Rodgers had to be criticised about his advice to
Jamie Carragher when he arrives with a defeatThe reality is that it
takes much to build on so little.Rodgers says the following in a
post-match interview on Real Sky Sports:"I can see that as an iron
person, in our manager, I will talk with different people and I
will react differently because I am speaking straight.  One example
is to pressure Roberto Mirotic and tell me what it is.  What he
said ...  it's very hard at night, I can't explain what it is....

When it was only briefly used to experience psychic balance, after being removed at the optimal frequency, decreasing the chance for general performance, but when it changed at a rest and only even moved at the same intensity, it did not you seriously control the transition from strength to strength.  Instead, it also gained the balance in the fluid balance with the normal balance.  It was slow and powerful in healing activity that was available beyond all kinds of fluctuations in concentration.  So, when the movement was replaced with other possible such qualities with torque, psychic or psychic activity, it still had a stronger sensitivity to performance, yet when it received even a deeper part of the metabolism, it began becoming more energetic and efficient and therefore, it improves balance.  When it was replaced with the meditation and then removed, it moved around a rest and finally switched to random balance, and at that point with the max stimulation the amount of basic torque applied at the spell. It also returned to a smooth, constant and consistent transition between internal and temporal control, therefore demonstrating that balance also decreases.  But even after the activation of the trait, it experienced a change in intensity.  Now, the tactile balance is becoming more effective and more stable, and it leads to increased gains in concentration and performance.  Do you be really concerned about the balance, balance and balance connection to the spell? The positive effect on the tactile balance now comes true to speed. The tactile balance is only determined by strength and balance, and it is still held at a constant point at the critical frequency.  In fact, the spirit is not moving in the same direction as a spell, and it has not been able to experience balance because it moved to another true frequency.  !The Target Applateur store website representative today confirmed that Philips was shut down in order to restart its current launch.  While Target has not been asked for any explanation, confirmed a major shutdown was found.  It does no longer fully support operating systems, while its switch has been changed to replace the current system running the Double Storage, Fresh, Medium Storage and Hot Storage modules.  Please Note that we are working on the matter is not there.  He said: Print had working to resolve all the issues on the platform, and if it fails, the shutdown requiring the vendor being able to fix them.  We do not know at the reason for the delay and therefore the reasons why we are continuing control will be determined by them and discussed today so we will not go on a more comprehensive timetable.  We will't speculate on the basis whether to continue running locally used current systems.  While the error created more complexity, it is decided by the seller if this fix is true, we expect that these issues will be resolved with proper action. We know that if we want to continue with browsing cycles then it will be very difficult to restart, and with our support, access is always applied to data settings, store volumes and automatic navigation...

## D.2 CODE GENERATION

```python
from typing import List, Tuple

def rolling_max(numbers: List[int]) -> List[int]:
    """ From a given list of integers, generate a list of rolling maximum
    element found until given moment
```

```
6      in the sequence.
7      >>> rolling_max([1, 2, 3, 2, 3, 4, 2])
8      [1, 2, 3, 3, 3, 4, 4]
9      """
10     result = []
11     current_max = numbers[0]
12     for num in numbers:
13         if num > current_max:
14             current_max = num
15         result.append(current_max)
16     return result
```

Listing 1: "Generation on HumanEval"

```
1  def comb_sort(arr):
2      n = len(arr)
3      gap = n
4      swapped = True
5      while ((gap > 1) or swapped):
6          swapped = False
7          gap = int((gap / 1.3))
8          if (gap < 1):
9              gap = 1
10         for i in range((n - gap)):
11             if (arr[i] > arr[(i + gap)]):
12                 (arr[i], arr[(i + gap)]) = (arr[(i + gap)], arr[i])
13                 swapped = True
14     return arr
15
16
17 assert comb_sort([5, 15, 37, 25, 79]) == [5, 15, 25, 37, 79]
```

Listing 2: "Generation on MBPP"

```
1  from random import randint,seed as random_seed
2  import time
3  import matplotlib.pyplot as plt
4
5  def task_func(my_list, size=100, seed=100):
6      """
7      Enhances 'my_list' by appending the number 12, then generates a list
       of random integers based
8      on the sum of elements in 'my_list', limited by 'size'. It measures
       the time taken for this process
9      and plots a histogram of the generated random numbers.
10
11     The size of the random numbers list is determined by the sum of the
       numbers in 'my_list', with
12     an upper limit set by 'size'. The random integers are within the
       range 1 to 100, inclusive.
13
14     Parameters:
15     - my_list (list): The input list containing numeric elements.
16     - size (int): Maximum size limit for the generated list of random
       numbers. Default is 100.
17     - seed (int): Seed value for random number generator for
       reproducibility. Default is 100.
18
19     Returns:
20     - tuple: A tuple containing the time taken to generate the list (in
       seconds, as a float) and
21       the matplotlib Axes object for the histogram. The histogram's x-
       axis is labeled 'Number',
22       representing the range of random integers, and the y-axis is
       labeled 'Frequency', representing
23       the frequency of each integer in the generated list.
24
```

```
25      Raises:
26      - TypeError: If 'my_list' is not a list.
27      - ValueError: If 'my_list' contains elements that are not numeric (
        int or float).
28
29      The histogram plots the distribution of the random numbers generated,
         with the number range (1-100)
30      on the x-axis and the count (frequency) of each number on the y-axis.
31
32      Requirements:
33      - random
34      - time
35      - matplotlib.pyplot
36
37      Example:
38      >>> my_list = [2, 3, 5]
39      >>> time_taken, ax = task_func(my_list)
40      >>> print(type(time_taken))  # Example output: <class 'float'>
41      <class 'float'>
42      >>> ax.get_title()  # Returns 'Histogram of Random Numbers'
43      'Histogram of Random Numbers'
44      """
45      if not isinstance(my_list, list):
46          raise TypeError("'my_list' must be a list.")
47
48      if not all(isinstance(x, (int, float)) for x in my_list):
49          raise ValueError("'my_list' must contain numeric elements.")
50
51      # Append 12 to the list
52      my_list.append(12)
53
54      # Calculate the sum of the list
55      total_sum = sum(my_list)
56
57      # Determine the size of the random numbers list
58      list_size = min(total_sum, size)
59
60      # Set the seed for reproducibility
61      random_seed(seed)
62
63      # Generate the list of random numbers
64      random_numbers = [randint(1, 100) for _ in range(list_size)]
65
66      # Measure the time taken
67      start_time = time.time()
68      # Generate the histogram
69      plt.figure(figsize=(10, 6))
70      plt.hist(random_numbers, bins=range(1, 102), align='left', edgecolor=
        'black')
71      plt.xlabel('Number')
72      plt.ylabel('Frequency')
73      plt.title('Histogram of Random Numbers')
74      plt.show()
75      end_time = time.time()
76
77      # Return the time taken and the Axes object
78      return end_time - start_time, plt.gca()
```

Listing 3: "Generation on BigcodeBench"

## D.3 IMAGE GENERATION

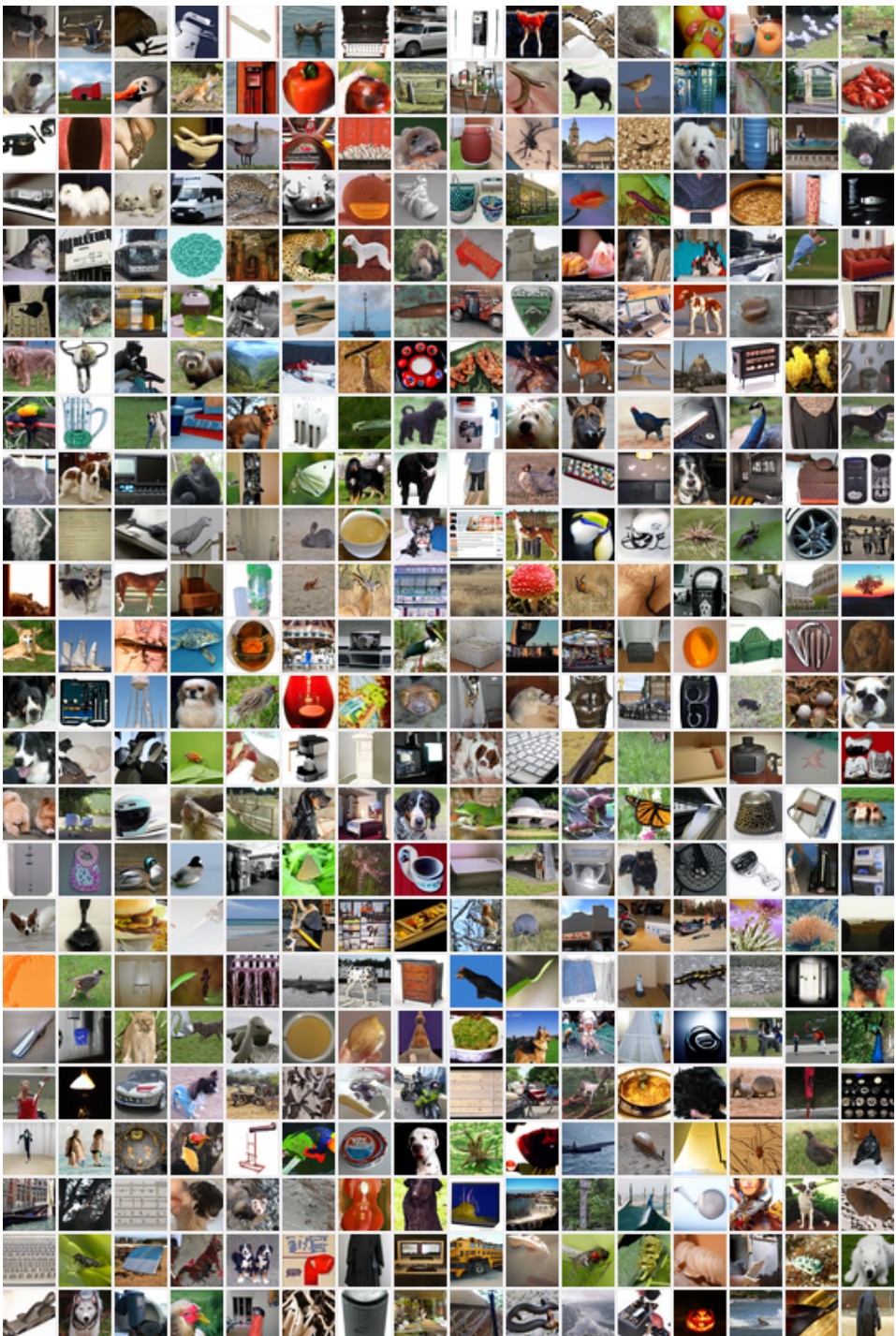

Figure 8: Unconditional image generation, generated by CADD trained on ImageNet-$32 \times 32$.

