# OpenReview forum: "Continuously Augmented Discrete Diffusion model for Categorical Generative Modeling"
_ICLR.cc/2026/Conference — ICLR 2026 Poster_

### Official Review · Reviewer_LxZ1 · 2025-10-30

**Soundness:** 4
**Presentation:** 3
**Contribution:** 3
**Rating:** 6
**Confidence:** 2

**Summary:**

This paper tackles a core problem in discrete diffusion models: all states collapse to a single absorbing state without retaining meaningful information. Inspired by the continuous diffusion model, the authors propose continuously augmented discrete diffusion (CADD), a hybrid framework that augments the standard masking process paired diffusion process in a continuous latent space. CADD utilizes a noisy continuous vector as a semantic condition to guide the discrete desnoising step. Experimental results across three distinct modalities demonstrate the effectiveness and versatility of CADD.

**Strengths:**

- The idea that combines the discrete diffusion process with the continuous semantic latent to resolve the information void problem is novel and seamlessly integrated into the training and sampling pipeline. Furthermore, the lower bound of the objective is well-justified.

- The paper's motivation and core concept are clearly presented, and the visualizations in Figures 1 and 2 are also clear.

- The effectiveness of CADD was validated across three distinct tasks, consistently demonstrating improvement over conventional discrete diffusion models. The paired continuous latent vector effectively resolves the 'information void' problem, which is clearly reflected in the scalable generation quality as sampling steps increase.

**Weaknesses:**

- While compatible with MDM training, the CADD framework is inherently more complex. It requires managing two parallel diffusion samplers and fusing their representations at every step.

- The authors opt to use only the standard discrete cross-entropy loss for simplicity, omitting the continuous MSE loss that would naturally arise from the ELBO. While this works well empirically, the paper could be strengthened by either (a) showing that adding the MSE loss provides no benefit (justifying its omission) or (b) exploring the full loss term, which might lead to even better performance.

- While CADD shows superior capabilities with large sampling steps, the performance seems to degrade sharply with a few-step sampling, probably due to the mode covering problem of the continuous semantic latent. This could amplify the inefficiency of the discrete diffusion models, which already suffer from large NFEs.

**Questions:**

- Modern visual generative models mostly utilize vector-quantized (VQ) visual tokens for image synthesis with complex conditioning (such as text prompts). As this paper primarily searches for unconditional visual generation with discrete pixel space, could CADD be impacted by the complex VQ tokens and conditioning?

- Masked Generative Models are a family of discrete diffusion models with an absorbing state, while they show superior efficiency [1][2]. Can the CADD framework be applied as a drop-in augmentation to MaskGIT-style models? If so, what is the expected trade-off in terms of sample quality/diversity?

[1] Maskgit: Masked generative image transformer
[2] An Image is Worth 32 Tokens for Reconstruction and Generation

**Details Of Ethics Concerns:**

No ethical concerns raised.

---

> ### Author Response · Authors · 2025-11-22
> **Reponse to Reviewer LxZ1**
>
> > **The CADD framework is inherently more complex. It requires managing two parallel diffusion samplers and fusing their representations at every step.**
>
> Please see our common response.
>
> > **The paper could be strengthened by either (a) showing that adding the MSE loss provides no benefit (justifying its omission) or (b) exploring the full loss term, which might lead to even better performance.**
>
> We add training results that include an extra MSE loss $|| \hat z_0 - z_0 ||^2$ using the two parameterizations of $\hat z_0$ (soft and hard) defined in Eq. 21. From the table below, CE + MSE provides performance that is close to using CE alone. The soft parameterization gives a gain under hard-inference MAUVE, but it introduces higher computation cost. This is because the soft prediction requires a matrix multiplication between the predicted probability vector $\mathbb{R}^{B \times L \times d}$ and the token embedding matrix $\mathbb{R}^{d \times V}$ with batch size $B$, sequence length $L$, embedding dimension $d$ and vocabulary size $V$. Such extra cost reduces TPS/GPU and increases TFLOPS. A possible explanation for the limited improvement is that, in the categorical setting, an MSE loss behaves similarly to cross entropy since both losses guide the model toward selecting the correct token and its embedding. We expect MSE to be more useful in settings where the targets are not purely categorical, and we leave the study of effective CE + MSE parameterizations to future work.
>
> | Metric               | CE        | CE + MSE (soft) | CE + MSE (hard) |
> |----------------------|-----------|------------------|------------------|
> | MAUVE (soft / hard)  | 0.24 / 0.18 | 0.24 / 0.24     | 0.22 / 0.24     |
> | TFLOPS               | 0.291     | 0.325           | 0.291           |
> | TPS/GPU              | 47,152    | 32,117          | 47,152          |
>
>
> > **The performance seems to degrade sharply with a few-step sampling, probably due to the mode covering problem of the continuous semantic latent. This could amplify the inefficiency of the discrete diffusion models, which already suffer from large NFEs.**
>
> We agree that current discrete diffusion models often require a large number of sampling steps to refine their mode coverage. After adding the continuous path in CADD, we already observe that with a small number of sampling steps, CADD performs better than pure discrete diffusion. This shows that the continuous path can improve sampling efficiency.
>
> There are also useful features of continuous diffusion that we have not fully explored in this work, such as more advanced solvers and score-distillation methods, which have been shown to reduce the required number of steps in continuous models. These techniques can likely make CADD-style models more efficient, and we are excited on these possible explorations and plan to study them in future work.
>
> > **As this paper primarily searches for unconditional visual generation with discrete pixel space, could CADD be impacted by the complex VQ tokens or conditioning?**
>
> Below we provide the conditional generation results on ImageNet-32. Compared to the unconditional generation, using smaller NFE (256, 512) can already surpass the performance of the unconditional  variant using NFE=1024, which makes generative modeling easier. Using guidance shows better performance than unguided cases, but possibly needs more careful tuning to achieve a better performance.
>
> | Steps         | 256    | 512   |
> |---------------|--------|-------|
> | cfg=1.5           | 2.82   | 2.64  |
> | cfg=1             | 3.14   | 3.08  |
> | unconditional (1024 steps)|  | 3.74   |
>
> > **Can the CADD framework be applied as a drop-in augmentation to MaskGIT-style models? If so, what is the expected trade-off in terms of sample quality/diversity?**
>
> We believe CADD can be applied as a drop-in augmentation to MaskGIT-style models, where the discrete branch of CADD is fully compatible with the iterative masking and prediction process used by MaskGIT. The continuous path can be added on top of the existing embedding pipeline without changing the decoder architecture.  In such design,
>
> * The continuous path provides a smooth refinement signal and encourages mode coverage. We expect diversity to remain stable or increase slightly.
> * The continuous path may allow the model to use fewer steps to reach a similar or better quality level.

---

> > ### Comment · Reviewer_LxZ1 · 2025-11-27
> >
> > Thank you for the detailed response. While the authors have addressed my concerns, the concern regarding efficiency remains (the performance decreases sharply as the sampling steps decrease). Given that, I keep my original score.

---

> > > ### Author Response · Authors · 2025-11-27
> > > **Thank you for the reply!**
> > >
> > > Dear Reviewer LxZ1,
> > >
> > > Thank you for the follow-up and confirming the addressed concerns. We understand the concerns when the number of sampling steps is small. We would like to clarify a few points.
> > >
> > > 1. **Reducing the number of sampling steps is not the main goal of CADD**, and we do not present low-step sampling as a contribution of this work. The focus is on coupling discrete and continuous paths within the proposed model to address issues in discrete (especially mask-based) diffusion and to improve performance, without adding computation cost in either training or sampling.
> > >
> > > 2. Even though low-step sampling is not our target, we still observe that CADD, through the inclusion of the continuous path, gives better performance than pure mask diffusion at smaller step counts. For example, in text generation, **CADD with 256 sampling steps already surpasses mask diffusion baselines (MDLM and SEDD) that use 4096 sampling steps**. This shows that CADD is substantially more efficient than pure mask diffusion in practice, even though reducing the number of steps is not the main goal here.
> > >
> > >
> > > 3. Following your suggestion, we tested CADD with class-conditioning and found that it can generate with fewer steps (256 or 512 steps) while maintaining stable performance, without the sharp degradation described in the concern.
> > >
> > > 4. We believe CADD may achieve even stronger efficiency with additional components. The continuous path offers features that we have not fully explored, such as more advanced solvers and score-distillation methods [1-5]. These techniques have been shown to reduce the number of steps required in continuous models, and they may further improve CADD, as we have already shown that the continuous variable $z_t$ (or equivalently $z_0$) conditioning plays a crucial role in unmasking. A complete study of these extensions is beyond the scope of this paper and is left for future work.
> > >
> > > We appreciate your time and assessment. Please let us know if you have any suggestions or remaining concerns.
> > >
> > >
> > > ------
> > >
> > > [1] Poole, Ben, Ajay Jain, Jonathan T. Barron, and Ben Mildenhall. "Dreamfusion: Text-to-3d using 2d diffusion." arXiv preprint arXiv:2209.14988 (2022).
> > >
> > > [2] Lu, Cheng, Yuhao Zhou, Fan Bao, Jianfei Chen, Chongxuan Li, and Jun Zhu. "Dpm-solver: A fast ode solver for diffusion probabilistic model sampling in around 10 steps." Advances in neural information processing systems 35 (2022): 5775-5787.
> > >
> > > [3] Song, Yang, Prafulla Dhariwal, Mark Chen, and Ilya Sutskever. "Consistency models." (2023).
> > >
> > > [4] Yin, Tianwei, Michaël Gharbi, Richard Zhang, Eli Shechtman, Fredo Durand, William T. Freeman, and Taesung Park. "One-step diffusion with distribution matching distillation." In Proceedings of the IEEE/CVF conference on computer vision and pattern recognition, pp. 6613-6623. 2024.
> > >
> > > [5] Zhou, Mingyuan, Huangjie Zheng, Yi Gu, Zhendong Wang, and Hai Huang. "Adversarial score identity distillation: Rapidly surpassing the teacher in one step." arXiv preprint arXiv:2410.14919 (2024).

---

### Official Review · Reviewer_xGtH · 2025-10-31

**Soundness:** 4
**Presentation:** 3
**Contribution:** 4
**Rating:** 6
**Confidence:** 3

**Summary:**

This paper is motivated by the information loss when transforming clean tokens into mask tokens in MDMs. To address this issue, CADD augments each token with an informative continuous latent vector, which can provide more information on the original tokens. I find this idea is quite novel and can be viewed as the most important contribution of this paper.
The authors discuss the training and sampling method of CADD and validate it on image, text, and code generation tasks.

**Strengths:**

- I find the idea of augmenting MDMs with continuous latent vectors to provide semantic information is both novel and well-motivated.
- The empirical evidence is strong to support the effectiveness of the CADD method.
- The paper presents a thorough derivation for training objective (ELBO) and sampling method, making the methodology rather practical.

**Weaknesses:**

- The authors use a learnable network $w_\theta$ which takes sequence $x$ as input and outputs the latent vector $z$. However, in my opinion, a more straightforward and robust way is directly assigning a learnable latent vector for each token, and all the parameters forms a "codebook". Will this parameterization works in practice?
- In line 5 of Algorithm 1, the $w_\theta$ takes the entire $x$ as input, and in line 10 of Algorithm 2, the $w_\theta$ takes a single token $x^i$ as input. Which one is the correct expression?
- The authors should discuss how they implement the  $w_\theta$ in practice. Moreover, as they introduce an additional model, a comparison on memory and computation time should be provided.
- I suggest that the authors to provide more experimental results. For example, the test perplexity and zero-shot perplexity can be a more  common benchmark for unconditional text generation. I am also interested in how the FID score on CIFAR10 scales on different choices of NFEs?

**Questions:**

- The current methodology of discrete diffusion almost converges to masked diffusion models. However, regarding the information void issue of MDMs, can this issue be alleviated by discrete diffusion with a uniform distribution noise?
- How do you compute your NFE? (e.g. in Algorithm 2)

---

> ### Author Response · Authors · 2025-11-22
> **Reponse to Reviewer xGtH**
>
> > **A more straightforward and robust way is directly assigning a learnable latent vector for each token, and all the parameters forms a "codebook". Will this parameterization works in practice?**
>
> To our understanding, the codebook you refer to is essentially the token embedding matrix $w_\theta$. Each token $x$ is already mapped to a latent vector through this codebook, that is, $z_0 = w_\theta(x_0)$. CADD builds a continuous diffusion process on top of this latent $z_0$ so that the continuous path stays aligned with the discrete diffusion steps that occur in the token space.
>
> If the codebook you refer to is time-dependent, meaning that each diffusion step t has its own learnable codebook, then the model must also define how these codebooks relate to each other across time. In that case, the diffusion process would also need an explicit transition rule that moves the latent from one time-dependent codebook to the next. This adds extra complexity, since the model must learn both the codebook at each step and the mapping between them. Without a clear transition structure, the model may have difficulty keeping the discrete and continuous paths consistent and reverse back from prior.
>
> Please let us know if our understanding to your question is correct and we are happy to discuss more.
>
> > **In line 5 of Algorithm 1, the takes the entire as input, and in line 10 of Algorithm 2, the takes a single token as input. Which one is the correct expression?**
>
> Both descriptions are correct. In line 5 of Algorithm 1, the embeddings at all positions are first initialized using the embedding matrix $w_\theta(x_t)$. We omit the position index $i$ as we operates on all positions. In line 7, the embeddings at the masked positions are then corrupted according to the continuous schedule. The unmasked positions remain unchanged.
>
> Similarly, in Algorithm 2, line 3 performs the same initialization for the full sequence, and the later steps apply the update rule based on whether each position is masked or not.
>
> > **A comparison on memory and computation time should be provided.**
>
> Please refer to the common response.
>
> > **Provide more experimental results. For example, the test perplexity and zero-shot perplexity can be a more common benchmark for unconditional text generation. I am also interested in how the FID score on CIFAR10 scales on different choices of NFEs?**
>
> We have provided the test perplexity and zero-shot perplexity in Table 6, 7 located in Appendix C.2, and the different choices of NFEs on Cifar-10 was in the table of Table 11 (Appendix C.3).
>
> > **Regarding the information void issue of MDMs, can this issue be alleviated by discrete diffusion with a uniform distribution noise?**
>
> Uniform diffusion shares key behaviors with continuous diffusion. As the corruption level increases, the data distribution becomes smoother and different modes move toward each other. This helps with enhancing mode coverage and reduces the risk of information voids. However, uniform diffusion does not provide strong mode-seeking behavior, which is the main advantage of mask diffusion. Some studies have shown that the mode-seeking properties are the key that mask diffusion has superior performance than uniform diffusion [1].
>
> CADD is designed to balance mode coverage and mode seeking. The continuous branch does not need to be tied to a single choice of Gaussian noise process, and other processes with similar properties could be used in place of the current continuous variable modeling. Uniform noise is a reasonable option to explore, and we will consider it in future work.
>
>
> [1] Amin, Alan N., Nate Gruver, and Andrew Gordon Wilson. "Why Masking Diffusion Works: Condition on the Jump Schedule for Improved Discrete Diffusion." arXiv preprint arXiv:2506.08316 (2025).
>
>
> > **How do you compute your NFE?**
>
> We count one model forward pass as one NFE. In other words, one sampling step corresponds to one NFE. Even though CADD has both a discrete path and a continuous path, the fused embedding allows the model to run only a single forward pass to produce the prediction at each step.

---

### Official Review · Reviewer_oUtt · 2025-10-31

**Soundness:** 3
**Presentation:** 3
**Contribution:** 3
**Rating:** 8
**Confidence:** 3

**Summary:**

The paper presents Continuously Augmented Discrete Diffusion (CADD), a framework which combines the strengths of Discrete Diffusion Models and Continuous Diffusion Models. CADD retains the discrete masking strategy and uses continuous latent vectors as the tokens get masked, to maintain semantic information. In essence, two diffusion processes happen at the same time. Experimental results show that CADD models consistently generate higher quality samples over the baselines in three tasks: text, image and code generation.

**Strengths:**

* The paper is well written, complete and easy to follow.
* It bridges a known gap and is conceptually novel. This specific combination of discrete diffusion with continuous latent vectors when the tokens are masked had not been previously explored.
* The idea is technically sound and the formulation of CADD is coherent. The factorization q(x_t​,z_t​ | x_0​)=q(x_t ​|x_0​)q(z_t​|x_t​,x_0​) provides a clear way to combine the discrete and continuous variables.
* There is strong empirical validation. Results are presented for 3 different tasks, and CADD offers significant improvement over the baselines.

**Weaknesses:**

* The discrete and continuous reverse process are derived in Section 4, but in the experiments the continuous MSE term is not used. This means that the continuous part is never directly trained to match the distribution, just guided. Although z_t appears to carry meaningful information, this makes me question how including the MSE would impact z_t and overall performance.
* CADD maintains an extra latent vector per masked token, when compared to standard Discrete Diffusion Models. The authors should discuss whether this results in extra computational or memory cost.
* In the text generation experiments, the authors do not compare CADD against any continuous/hybrid diffusion model, such as Duo.

**Questions:**

1. What happens if you include the MSE loss? How does this affect z_t and performance?
2. Does CADD incur extra computational cost vs discrete diffusion models? If so, is it significant?
3. Did the authors test any alternative schedules for the continuous process?
4. In the main experiments for text generation, why isn’t Duo a baseline?
5. It could be interesting to include interpretability analysis regarding z_t, to visualize examples where the latent hints help disambiguate the tokens.

---

> ### Author Response · Authors · 2025-11-22
> **Reponse to Reviewer oUtt (part 1)**
>
> > **How including the MSE would impact z_t and overall performance**
>
> We add training results that include an extra MSE loss $|| \hat z_0 - z_0 ||^2$ using the two parameterizations of $\hat z_0$ (soft and hard) defined in Eq. 21. From the table below, CE + MSE provides performance that is close to using CE alone. The soft parameterization gives a gain under hard-inference MAUVE, but it introduces higher computation cost. This is because the soft prediction requires a matrix multiplication between the predicted probability vector $\mathbb{R}^{B \times L \times d}$ and the token embedding matrix $\mathbb{R}^{d \times V}$ with batch size $B$, sequence length $L$, embedding dimension $d$ and vocabulary size $V$. Such extra cost reduces TPS/GPU and increases TFLOPS. A possible explanation for the limited improvement is that, in the categorical setting, an MSE loss behaves similarly to cross entropy since both losses guide the model toward selecting the correct token and its embedding. We expect MSE to be more useful in settings where the targets are not purely categorical, and we leave the study of effective CE + MSE parameterizations to future work.
>
> | Metric               | CE        | CE + MSE (soft) | CE + MSE (hard) |
> |----------------------|-----------|------------------|------------------|
> | MAUVE (soft / hard)  | 0.24 / 0.18 | 0.24 / 0.24     | 0.22 / 0.24     |
> | TFLOPS               | 0.291     | 0.325           | 0.291           |
> | TPS/GPU              | 47,152    | 32,117          | 47,152          |
>
> > **CADD maintains an extra latent vector per masked token, when compared to standard Discrete Diffusion Models. The authors should discuss whether this results in extra computational or memory cost**
>
> Please see our common response.
>
> > **The authors do not compare CADD against any continuous/hybrid diffusion model, such as Duo**
>
> We compared the test perplexities and zero-shot perplexities with Duo in Appendix Table 6 and Table 7. Below we add the downstream MAUVE comparison with Duo and have updated the Figure 4 in the main text. From the results, Duo shows better performance than MDLMs while still having a gap to CADD when number of sampling steps goes beyond 128. Duo with consistency distillation can reach performance that is close to CADD. This supports the view that score-distillation methods can guide the model toward stronger mode-seeking behavior while still keeping reasonable mode coverage, which leads to improved downstream quality.
>
> Another observation is that when the number of sampling steps becomes larger than the number of tokens to generate, Duo and Duo-DCD show a clear performance drop, while CADD continues to improve in this regime. This aligns with the intended design of CADD, where the continuous branch and the discrete branch reinforce each other even when the number of refinement steps is high.
>
> | Method   | 128    | 256    | 512    | 1024   | 4096   |
> |----------|--------|--------|--------|--------|--------|
> | CADD     | 0.017  | 0.084  | 0.141  | 0.244  | 0.272  |
> | Duo      | 0.063  | 0.062  | 0.062  | 0.117  | 0.087  |
> | Duo-DCD  | 0.096  | 0.080  | 0.137  | 0.240  | 0.128  |

---

> ### Author Response · Authors · 2025-11-22
> **Reponse to Reviewer oUtt (part 2)**
>
> > **Did the authors test any alternative schedules for the continuous process?**
>
> We add the Variance-Preserving (VP) schedule to both text and image generation experiments. For text generation, the results are on par with the Flow-matching schedule. For image generation, the FiD score is slightly worse under the VP schedule. This difference may come from the need for different hyper-parameter settings for the two schedules.
>
> |	|FM	|VP |
> |----------|--------|--------|
> |MAUVE |	0.2403	|0.2403|
> |FiD	|2.88| 3.16|
>
> > **Interpretability analysis regarding z_t, to visualize examples where the latent hints help disambiguate the tokens**
>
> We add an illustrative example in Table 12 of the revised pdf (listed below). To demonstrates the effect of the continuous embedding $z_t$ as a semantic scaffold that guides the discrete unmasking process, we use "*A model sits on the mat*" as input, and mask the second position. Using different noise level $t$ to corrupt the embedding of token "*model*" to form $z_t$ and predict the masked token. In the low-noise regime ($t \le 0.3$), the continuous signal is clear enough and the prediction is same as "*model*". As noise increases, the embedding degrades and we can observe the predictions traverse from relevant category (e.g., "*vase*", "*tank*") to generic priors only when the guiding signal is fully destroyed at $t=1.0$.
>
> | Noise Level (t) | Predicted Sentence            | Predicted Token |
> |-----------------|--------------------------------|------------------|
> | Input (GT)  | A **model** sits on the mat        | model           |
> | t = 0.1         | A **model** sits on the mat        | model           |
> | t = 0.3         | A **model** sits on the mat        | model           |
> | t = 0.5         | A **vase** sits on the mat         | vase            |
> | t = 0.7         | A **~tank** sits on the mat         | ~tank            |
> | t = 1.0         | A **and** sits on the mat          | and             |

---

### Official Review · Reviewer_7PKE · 2025-11-01

**Soundness:** 3
**Presentation:** 3
**Contribution:** 3
**Rating:** 6
**Confidence:** 4

**Summary:**

This paper introduces Continuously Augmented Discrete Diffusion (CADD), a framework that combines Masked Discrete Diffusion (MDM) with a parallel continuous diffusion process in the embedding space. The key innovation is to replace the "information void" of a [MASK] token with a noisy but semantically informative continuous latent vector. This provides "soft hints" during the denoising process, leading to more coherent and higher-quality generation across text, image, and code modalities. The method is shown to be effective, scalable, and simple to implement.

**Strengths:**

The paper target at solving a limitation of existing MDMs, which is the "information void" created by the absorbing [MASK] state. The idea of augmenting the discrete space with a continuous one to preserve graded semantic information is both intuitive and powerful.

The paper provides extensive experiments across multiple modalities (text, image, code) and scales (from 168M to 7B parameters). Consistent gains are observed over baselines.

The paper is well-written.

**Weaknesses:**

Time cost of model running can be analyzed. For example, a time cost and FLOPs comparison between CADD and the baselines can be helpful for the readers to understand the performance-efficiency trade-off.

The experiment comparison is presented mainly with statistical result. It lacks illustrative example to show how the latent space parameter helps improve the generation performance.

Code is not released for reproducibility checking.

**Questions:**

What is the inference-time latency of CADD (with K=1) compared to a standard MDM baseline when generating a sequence of fixed length? Is the overhead primarily from the extra continuous state operations or from the architectural changes?

The continuous latent z_t is initialized from a standard Gaussian prior at step T. Have you experimented with more informative priors, and if so, what was the effect?

---

> ### Author Response · Authors · 2025-11-22
> **Reponse to Reviewer 7PKE**
>
> > **Time cost and FLOPs of model running can be analyzed + What is the inference-time latency of CADD (with K=1) compared to a standard MDM baseline when generating a sequence of fixed length?**
>
> Please refer to our responses 1 to all reviewers.
>
>
> > **The experiment comparison is presented mainly with statistical result. It lacks illustrative example to show how the latent space parameter helps improve the generation performance.**
>
> We add an illustrative example in Table 12 of the revised pdf (listed below). To demonstrates the effect of the continuous embedding $z_t$ as a semantic scaffold that guides the discrete unmasking process, we use "*A model sits on the mat*" as input, and mask the second position. Using different noise level $t$ to corrupt the embedding of token "*model*" to form $z_t$ and predict the masked token. In the low-noise regime ($t \le 0.3$), the continuous signal is clear enough and the prediction is same as "*model*". As noise increases, the embedding degrades and we can observe the predictions traverse from relevant category (e.g., "*vase*", "*tank*") to generic priors only when the guiding signal is fully destroyed at $t=1.0$.
>
> | Noise Level (t) | Predicted Sentence            | Predicted Token |
> |-----------------|--------------------------------|------------------|
> | Input (GT)  | A **model** sits on the mat        | model           |
> | t = 0.1         | A **model** sits on the mat        | model           |
> | t = 0.3         | A **model** sits on the mat        | model           |
> | t = 0.5         | A **vase** sits on the mat         | vase            |
> | t = 0.7         | A **~tank** sits on the mat         | ~tank            |
> | t = 1.0         | A **and** sits on the mat          | and             |
>
>
> > **Code is not released for reproducibility checking.**
>
> We have provided full experiment settings and clear algorithm descriptions in both main text and appendices so that the procedure can be followed without hidden steps. We will release the code upon acceptance, which will allow more complete reproducibility.
>
> > **The continuous latent z_t is initialized from a standard Gaussian prior at step T. Have you experimented with more informative priors, and if so, what was the effect?**
>
> In this work we use a standard Gaussian prior $z_T \sim \mathcal{N}(0, I)$ without loss of generality. We did not explore more structured or data-dependent priors in a systematic way, because our focus was on the discrete–continuous coupling and the CADD training objective. More informative or learned priors (such as using pretrained model to provide diffusion starting point [1-3]) are interesting and compatible with CADD, and we leave a careful study of such designs to future work.
>
> -------
> *Reference*:
>
> [1] Pandey, K., Mukherjee, A., Rai, P., & Kumar, A. (2022). Diffusevae: Efficient, controllable and high-fidelity generation from low-dimensional latents. arXiv preprint arXiv:2201.00308.
>
> [2] Zheng, H., He, P., Chen, W., & Zhou, M. (2022). Truncated Diffusion Probabilistic Models and Diffusion-based Adversarial Auto-Encoders. arXiv preprint arXiv:2202.09671.
>
> [3] Lyu, Z., Xu, X., Yang, C., Lin, D., & Dai, B. (2022). Accelerating diffusion models via early stop of the diffusion process. arXiv preprint arXiv:2205.12524.

---

### Author Response · Authors · 2025-11-22
**Common response to all reviewers**

We thank all reviewers for the helpful feedback and suggestions. Below we first provide a common response that addresses the computation cost analysis and summarizes the revisions we have made. We then provide point-to-point responses to each concern and question raised by the individual reviewers.


> **The computation cost and FLOPs analysis**

We measured both training and inference efficiency under the same hardware (H100, FP32) and batch settings (batch size=1), using the DiT model with 169 M parameters. The results are summarized in the table. During training, CADD matches the speed of the MDM baseline, with nearly identical token throughput (tokens/s) per gpu and memory usage. During inference, when K = 1, CADD shows similar computation cost to MDM. The inference cost increases in a linear way as K increases, since the model performs K times forward for the $z_0$ estimation, while the training cost remains unchanged.

| Setting     | Token/s/GPU | TFLOPS | TFLOPS/s | Avg GPU Mem (GB) |
|-------------|-------------|--------|----------|-------------------|
| **Training** |             |        |          |                   |
| mdlm        | 47,416      | 0.29   | 6.049    | 1.362             |
| cadd        | 47,152      | 0.291  | 6.082    | 1.492             |
| **Inference** |           |        |          |                   |
| k = 1       | 47,373      | 0.291  | 6.082    | 1.492             |
| k = 2       | 24,852      | 0.581  | 12.206   | 3.511             |
| k = 3       | 15,768      | 0.873  | 18.310   | 5.253             |
| k = 4       | 1,417       | 1.162  | 24.417   | 7.470             |


> **Summary of revisions (all revised parts are marked in red to make the changes easy to check)**

* Computation cost analysis:  Table 4 added to Appendix B.3  (page 23)
* Duo results added to comparison: Update Figure 3 in Main experiments section 5.1 (page 8)
* Ablation on continuous schedule: Table 9  in Appendix C.3 (page 25)
* The use of CE, CE+MSE for training and the estimation of $\hat z_0$: Table 10 in Appendix C.3 (page 25)
* Visualization of the role of continuous embeddings: Table 12 in Appendix C.3 (page 25)

---

### Author Response · Authors · 2025-12-03
**Summary of Rebuttal**

Dear AC and reviewers,

We thank the reviewers for their time in careful reading and providing helpful feedback. To ease the review process, we prepared the following summary for the AC.

All reviewers initially provided positive scores (6/6/8/6) before the rebuttal started. The main requested changes were:
1.    clarification on computation cost and sampling efficiency compared with discrete diffusion models (all reviewers);
2.    qualitative examples to illustrate the role of the continuous path (Reviewer 7PKE and Reviewer oUtt);
3.    ablation of the choice of using MSE loss combining with CE loss (Reviewer oUtt and Reviewer LxZ1);
4.    clarifications and additional experiments such as conditional image generation (Reviewer xGtH and Reviewer LxZ1).

------

Below is a concise summary of our responses and revisions.

1. **Computation cost and sampling efficiency**

We provide full training and inference measurements (added as Table 4 in Appendix B.3, page 23). The conclusion is that CADD has the same computation cost as mask diffusion (MDLM/SEDD) in both training and sampling when using the same number of steps. CADD only introduces extra cost when drawing multiple continuous latents, which is an optional setting used to boost performance.

Regarding sampling efficiency, although reducing the number of sampling steps is not a goal of this work, CADD already outperforms MDLM and SEDD at much smaller step counts. For example, in text generation CADD with 256 steps surpasses MDLM and SEDD with 4096 steps, showing that CADD is more efficient in practice despite not being designed specifically for step reduction.

2. **Qualitative example**


We add an illustrative example in Table 12 of the revised manuscript. The example uses embeddings at different noise levels to show how the continuous latent influences the final output token. High-noise latents introduce more stochasticity, giving more diverse outputs, while low-noise latents produce more certain guidance. This example supports that CADD uses the continuous path to avoid the information void when a token is masked.


3. **Choice of loss parameterization**


We add an ablation comparing CE, CE + MSE, and the two parameterizations of continuous latents for MSE loss calculation in this setting (Table 10 in Appendix C.3, page 25). The results show that CE + MSE gives similar performance to CE alone, and CE remains preferable in the categorical setting due to lower computation cost. We explain that, for categorical targets, an MSE loss behaves similarly to cross entropy because both losses guide the model to select the correct token and its embedding. MSE may be more beneficial in broader settings where the targets are not strictly categorical, and we leave that investigation to future work.


4. **Additional experiments**


We add class-conditional image generation results. With guidance, CADD maintains strong performance at 256–512 steps, without the sharp degradation noted in the concern. We also include additonal VP-schedule results, unconditional vs. conditional comparisons, Duo baselines etc.

With these new results, figures, tables, and explanations have been updated accordingly. All revised content is marked in red in the updated manuscript.

---

### Meta-Review · Area_Chair_WGhJ · 2026-01-06

**Summary:**

The reviewers were broadly positive about the paper’s motivation, technical formulation, and empirical results. The main concerns raised focused on (i) computational cost and sampling efficiency relative to standard discrete diffusion models, (ii) clarification of the role and necessity of the continuous latent and associated loss terms, and (iii) requests for additional qualitative examples or ablations to better illustrate the benefits of the continuous augmentation.

In the rebuttal, the authors addressed these points with added efficiency measurements, ablations comparing CE vs. CE+MSE objectives, additional baseline comparisons, and illustrative qualitative examples demonstrating how the continuous latent provides semantic guidance during denoising. These responses satisfactorily resolved the reviewers’ questions, and no substantive technical concerns remain. Overall, the feedback supports acceptance.

**Reviewer Concerns:**

**Concerns largely addressed in the rebuttal**

- **Computational cost and sampling efficiency** (Reviewers **7PKE**, **oUtt**, **xGtH**, **LxZ1**):
  The authors added detailed training and inference cost measurements (Table 4, Appendix B.3), showing that CADD has comparable cost to MDM baselines when using K=1, with linear overhead only when multiple continuous samples are used. They also clarified that CADD already outperforms discrete baselines at substantially fewer steps in practice.

- **Role and necessity of the continuous latent / loss design** (Reviewers **oUtt**, **LxZ1**):
  Ablations comparing CE vs. CE+MSE (Table 10, Appendix C.3) demonstrate similar performance, justifying the use of CE alone for efficiency. Additional explanations clarify why MSE offers limited benefit in strictly categorical settings.

- **Qualitative and interpretability evidence for the continuous path** (Reviewers **7PKE**, **oUtt**):
  The authors added illustrative qualitative examples (Table 12) showing how different noise levels in the continuous latent affect token predictions, directly supporting the claimed semantic guidance effect.

- **Baseline coverage and additional experiments** (Reviewers **oUtt**, **xGtH**, **LxZ1**):
  Comparisons with Duo/Duo-DCD, alternative continuous schedules (VP vs. FM), test and zero-shot perplexity, conditional image generation, and additional NFE analyses were added, addressing the requested experimental gaps.

**Remaining limitations**

- **Few-step sampling efficiency** (Reviewer **LxZ1**):
  Reviewer LxZ1 noted that CADD is less competitive at very low sampling steps. The rebuttal clarified that few-step sampling efficiency is not a primary goal of the paper, and provided evidence that CADD still improves over pure masked diffusion models even in low-step regimes, as well as additional conditional generation results where fewer steps suffice. This point reflects a scope limitation rather than a technical deficiency, and does not affect the correctness or effectiveness of the proposed method in its intended setting.

Overall, the rebuttal adequately addressed the substantive technical and empirical concerns raised by the reviewers, with only a minor, explicitly acknowledged limitation remaining.

**Reviewer Scores:**

- **Reviewer 7PKE**: Initial score 6. Given that their main concerns (computational cost analysis and qualitative illustration of the continuous latent) were fully addressed in the rebuttal, it is reasonable to expect a modest increase, likely to **7 (possibly 8)**.

- **Reviewer oUtt**: Initial score 8. The reviewer’s questions regarding loss design, baselines (e.g., Duo), schedules, and interpretability were all addressed with additional experiments and clarifications. The score would likely **remain at 8**.

- **Reviewer xGtH**: Initial score 6. The rebuttal addressed the raised concerns on implementation details, baselines, and additional evaluations. While no explicit post-rebuttal score update was provided, a **small positive adjustment to 7 is plausible**, though conservatively the score may remain **around 6**.

- **Reviewer LxZ1**: Initial score 6. Despite acknowledging that most concerns were addressed, the reviewer explicitly stated that the concern regarding few-step sampling efficiency remains and that they **kept their original score**.

Overall, based on explicit post-rebuttal comments and reasonable interpretation where no updates were provided, a conservative post-discussion score profile is approximately **7, 8, 6, and 6**, corresponding to a **clear accept** with strength appropriate for a **poster presentation**.

---

### Decision · Program_Chairs · 2026-01-26

Accept (Poster)